# Nascent RHOH acts as a molecular brake on actomyosin-mediated effector functions of inflammatory neutrophils

Shuang Peng[1], Darko Stojkov[1], Jian Gao[2], Kevin Oberson[1], Philipp Latzin[3], Carmen Casaulta[3], Shida Yousefi[1], Hans-Uwe Simon [1,4,5,6]*

**1** Institute of Pharmacology, University of Bern, Bern, Switzerland, **2** State Key Laboratory of Pharmaceutical Biotechnology, Department of Biotechnology and Pharmaceutical Sciences, School of Life Sciences, Nanjing University, Nanjing, China, **3** Division of Respiratory Medicine, Department of Pediatrics, University Children's Hospital of Bern, University of Bern, Bern, Switzerland, **4** Department of Clinical Immunology and Allergology, Sechenov University, Moscow, Russia, **5** Laboratory of Molecular Immunology, Institute of Fundamental Medicine and Biology, Kazan Federal University, Kazan, Russia, **6** Institute of Biochemistry, Brandenburg Medical School, Neuruppin, Germany

* hus@pki.unibe.ch

**Data Availability Statement:** 'The re-analyzed public accessible single-cell RNA sequencing (scRNA-seq) datasets can be found at: https://www.ncbi.nlm.nih.gov/geo/query/acc.cgi?acc=

## Abstract

In contrast to molecular changes associated with increased inflammatory responses, little is known about intracellular counter-regulatory mechanisms that control signaling cascades associated with functional responses of neutrophils. Active RHO GTPases are typically considered as effector proteins that elicit cellular responses. Strikingly, we show here that RHOH, although being constitutively GTP-bound, limits neutrophil degranulation and the formation of neutrophil extracellular traps (NETs). Mechanistically, RHOH is induced under inflammatory conditions and binds to non-muscle myosin heavy chain IIA (NMHC IIA) in activated neutrophils in order to inhibit the transport of mitochondria and granules along actin filaments, which is partially reverted upon disruption of the interaction with NMHC IIA by introducing a mutation in RhoH at lysine 34 (RhoH$^{K34A}$). In parallel, RHOH inhibits actin polymerization presumably by modulating RAC1 activity. In vivo studies using Rhoh$^{-/-}$ mice, demonstrate an increased antibacterial defense capability against Escherichia coli (E. coli). Collectively, our data reveal a previously undefined role of RHOH as a molecular brake for actomyosin-mediated neutrophil effector functions, which represents an intracellular regulatory axis involved in controlling the strength of an antibacterial inflammatory response.

## Introduction

Neutrophils are versatile innate immune cells, playing central roles not only in the defense against invading pathogens but also in the regulation of innate as well as adaptive immune responses [1–3]. Neutrophils have long been considered as terminally differentiated cells with limited transcriptional activity, while a growing body of studies has recently described neutrophil transcriptional reprogramming induced by immune complexes, cytokines and danger- or

GSE158055; https://www.ebi.ac.uk/biostudies/arrayexpress/studies/E-MTAB-8832. The raw FCS files are available at the FlowRepository (https://flowrepository.org/) under ID: FR-FCM-Z5MZ, FR-FCM-Z5LV, FR-FCM-Z5M3, FR-FCM-Z5MY, FR-FCM-Z5M2. Other data can be found in the Supporting information files.'

**Funding:** This work was supported by the Swiss National Science Foundation (grant No. 310030_184816 to HUS and 31003A_173215 to SY) and the Russian Government Program "Recruitment of the Leading Scientists into the Russian Institutions of Higher Education" (grant No. 075-15-2021-600 to HUS). The funders had no role in study design, data collection and analysis, decision to publish, or preparation of the manuscript.

**Competing interests:** The authors have declared that no competing interests exist.

**Abbreviations:** C5a, complement component 5a; CEB, cytosolic extraction buffer; CF, cystic fibrosis; CFTR, cystic fibrosis transmembrane conductance regulator; CFU, colony-forming unit; Co-IP, co-immunoprecipitation; COVID-19, Coronavirus Disease 2019; dsDNA, double-stranded DNA; EV, empty vector; G-CSF, granulocyte colony-stimulating factor; GM-CSF, granulocyte/macrophage colony-stimulating factor; HBSS, Hanks' balanced salt solution; HD, healthy donor; HPC, hematopoietic progenitor cell; LPS, lipopolysaccharide; MLCK, myosin light chain kinase; mtDNA, mitochondrial DNA; NE, neutrophil elastase; NET, neutrophil extracellular trap; NK, natural killer; NMHC IIA, non-muscle myosin heavy chain IIA; PI, propidium iodide; PLF, peritoneal lavage fluid; PNL, postnuclear lysate; RLC, regulatory light chain; SCF, stem cell factor; scRNA-seq, single-cell RNA sequencing; SPF, specific-pathogen-free; WT, wild-type.

pathogen-associated molecular patterns in a variety of inflammatory contexts [4–9], in which neutrophil phenotypic and functional plasticity have been related to disease outcomes. There is increasing interest in selective modulation of neutrophil responses by targeting neutrophil transcriptional alterations in diseases [4–6,10,11]. However, it remains poorly understood how neutrophil transcriptional adaptations instruct their effector functions during inflammation.

Cystic fibrosis (CF) is a life-threatening genetic disease caused by mutations in the gene encoding cystic fibrosis transmembrane conductance regulator (CFTR) [12]. Progressive lung disease, characterized by buildup of mucus in the airways, recurrent bacterial infection, and sustained neutrophilic inflammation, is the dominant clinical manifestation and the leading cause of morbidity and mortality in patients with CF [12]. There is evidence that transcriptional firing represses neutrophil canonical functions in CF [13]. Nevertheless, in-depth studies are required to elucidate the molecular mechanisms underlying adaptive changes in CF neutrophils.

RHOH is a member of the RHO GTPase family, which orchestrates a wide range of cellular processes, such as cytoskeletal reorganization, vesicle trafficking, cell cycle, cell polarity, and cell migration [14–16]. Unlike typical RHO GTPases that undergo cycling between inactive GDP-bound and active GTP-bound conformation, RHOH lacks intrinsic GTPase activity and remains in a constitutively GTP-bound form [17]. The cellular activity of RHOH therefore appears to depend on its expression level and/or posttranslational modifications. RHOH exhibits high expression in T cells and plays crucial roles in T cell development, activation and differentiation [18–21], and undergoes degradation in lysosomes following TCR stimulation [22]. In addition, RhoH has been found to regulate proliferation, survival, migration and engraftment of hematopoietic progenitor cells (HPCs) [23], IL-3 signaling in a B cell line [24], FcεRI-dependent signal transduction in mast cells [25], and differentiation in eosinophils [26]. Previously, neutrophils isolated from CF patients showed increased protein expression of RHOH compared to neutrophils from healthy donors [27]. Up-regulated *RHOH* mRNA expression was observed in human neutrophils following stimulation with granulocyte/macrophage colony-stimulating factor (GM-CSF) [28]. The impact of aberrant RHOH expression in neutrophil functions, however, is largely unknown.

Neutrophils rely heavily on the cytoskeleton to respond to the local microenvironment and perform effector functions [29]. For instance, activated neutrophils rapidly form extracellular traps (NETs), which are web-like structures with enriched granule proteins assembled on a mitochondrial DNA (mtDNA) scaffold [30–36]. Actin and microtubule cytoskeleton-mediated transport of mitochondria and granules toward cell membrane is a prerequisite for the release of mtDNA and granule proteins into the extracellular space [35,36]. Intracellular cargo transport is typically driven by molecular motors traveling along the constructed cytoskeletal tracks, including myosins along actin filaments and kinesin and cytoplasmic dyneins on microtubules [37].

In this study, we demonstrate that neutrophils display increased RHOH expression after exposure to inflammatory stimuli and concomitant decreased ability in degranulation and NET formation. We further show that RHOH acts as a molecular brake during actin cytoskeleton-mediated intracellular transport of granules and mitochondria. Specifically, RhoH interacts with actin motor protein NMHC IIA upon neutrophil activation and decreases the ability of NMHC IIA in binding F-actin. In parallel, RhoH suppresses actin rearrangements by inhibiting Rac1 activity. In vivo studies using *Rhoh*<sup>-/-</sup> mice demonstrate an increased antibacterial defense capability against *Escherichia coli*. Together, our data reveal a novel regulatory axis involving RHOH and the actomyosin transport machinery, by which neutrophils respond to extrinsic inflammatory signals and control downstream effector functions.

## Results

### Extrinsic inflammatory signals induce RHOH expression in human neutrophils associated with reduced degranulation and NET formation

*RHOH* expression is hardly detectable in freshly isolated human and mouse neutrophils; however, its expression is up-regulated by GM-CSF stimulation in less than 3 h (S1A Fig), confirming previously published work [27,28,38]. High expression of *RHOH* is also detected in neutrophils from Coronavirus Disease 2019 (COVID-19) patients [39] and in tumor-infiltrating neutrophils in mice [40] as assessed by analyzing publicly accessible single-cell RNA sequencing (scRNA-seq) datasets (GEO GSE158055 and ArrayExpress E-MTAB-8832, respectively) (S1B and S1C Fig), suggesting that RHOH represents a shared transcriptional adaptation conferred to neutrophils by inflammatory microenvironments in distinct neutrophilic diseases.

To assess the impact of RHOH in neutrophil effector function, we isolated blood neutrophils from healthy donors (HD) and CF patients with signs of systemic inflammation (Fig 1A) and confirmed the previously reported high RHOH expression in CF neutrophils (Fig 1B) [27]. There was no obvious difference in ROS production between activated HD and CF neutrophils, as assessed by flow cytometry (Fig 1C). However, degranulation was impaired in CF neutrophils compared to HD neutrophils, as shown by reduced cell surface expression of CD63 and CD66b (surrogate marker for azurophilic granules and specific granules, respectively) and elastase activity (Fig 1A and 1D). Moreover, neutrophils isolated from CF patients failed to form detectable extracellular DNA fibers upon activation (Fig 1F). Consistently, we observed a significant decrease in the amount of double-stranded DNA (dsDNA) release in the supernatants of activated CF neutrophils relative to HD neutrophils (Fig 1G). As a consequence, CF neutrophils generated no detectable NETs, as demonstrated by colocalization of extracellular DNA fibers and neutrophil elastase (NE) (Fig 1H).

To confirm that the observed reduced functionality of CF neutrophils was secondary owing to the inflammatory microenvironment, we incubated human neutrophils from HDs with serum from CF patients. In contrast to HD serum, CF serum significantly elevated the protein level of RHOH (S2A Fig). Moreover, following activation, neutrophils treated with CF serum exhibited a dramatic decrease in degranulation (S2B Fig), dsDNA release (S2C Fig), and in the formation of NETs (S2D Fig), as compared with neutrophils treated with HD serum. These transcriptional and functional changes, which mirrored the phenotype of CF neutrophils, pointed to the hypothesis that inflammatory signals regulate neutrophil effector functions with an intracellular pathway that involves RHOH.

### Induction of RhoH expression by GM-CSF stimulation or overexpression of HA-tagged RhoH restricts mouse neutrophil effector functions

Considering the heterogeneity of CF neutrophils due to disease severity and treatments, we further investigated the functional role of RhoH in purified neutrophils from bone marrow of wild-type (WT) and *Rhoh*[-/-] mice (Fig 2A, S3A and S3B Fig). *Rhoh*[-/-] neutrophils were able to degranulate (Fig 2B) and extrude their dsDNA (Fig 2C and 2D) as efficiently as WT neutrophils in response to C5a following GM-CSF priming. Similar to human neutrophils, primary WT mouse neutrophils expressed barely detectable RhoH protein but up-regulated their RhoH expression following GM-CSF stimulation for 3 h (Fig 2A). Intriguingly, the 3 h stimulated WT neutrophils displayed impaired degranulation (Fig 2B) and dsDNA release (Fig 2C and 2D), while *Rhoh*[-/-] neutrophils treated equally did not show such changes (Fig 2A–D).

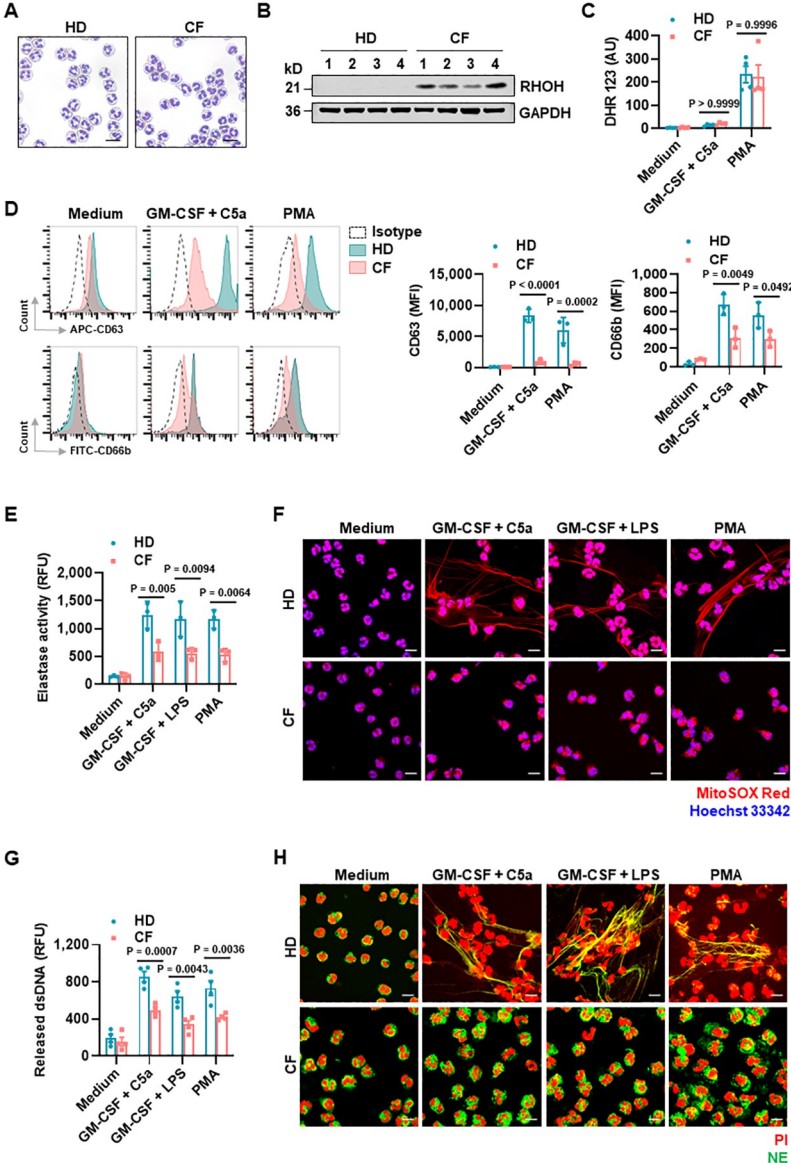

**Fig 1. Neutrophils from CF patients exhibit enhanced expression of RHOH associated with reduced degranulation and NET formation.** (**A**) Representative images of neutrophils from HD and CF patients (*n* = 4). (**B**) RHOH expression in neutrophils from HD and CF patients were analyzed by immunoblotting. (**C–H**) Freshly isolated neutrophils from HD and CF patients were treated with the indicated stimuli. (**C**) ROS activity was assessed by flow cytometry (*n* = 4). (**D**) Neutrophil degranulation was determined by flow cytometry (*n* = 3, sample number: 1, 2, 3). (**E**) Quantification of released NE in the culture supernatants (*n* = 3, sample number: 1, 2, 3). (**F**) Extracellular DNA fibers were stained with MitoSOX Red and the nucleus with Hoechst 33342 and analyzed by confocal microscopy (*n* = 3, sample number: 1, 2, 3). Scale bars, 10 μm. (**G**) Quantification of released dsDNA in the culture supernatants (*n* = 4). (**H**) NET formation indicated by the colocalization of NE (green) with released DNA (PI, red) was analyzed by confocal microscopy (*n* = 3, sample number: 1, 2, 4). Scale bars, 10 μm. Values are means ± SD. Two-way ANOVA with Tukey's multiple comparison post-test was applied. The underlying data for Fig 1C–E and 1G can be found in S1 Data. The underlying data for Fig 1B can be found in S1 Raw images. AU, arbitrary units; CF, cystic fibrosis; dsDNA, double-stranded DNA; MFI, mean fluorescence intensity; NET, neutrophil extracellular trap; PI, propidium iodide; RFU, relative fluorescence units.

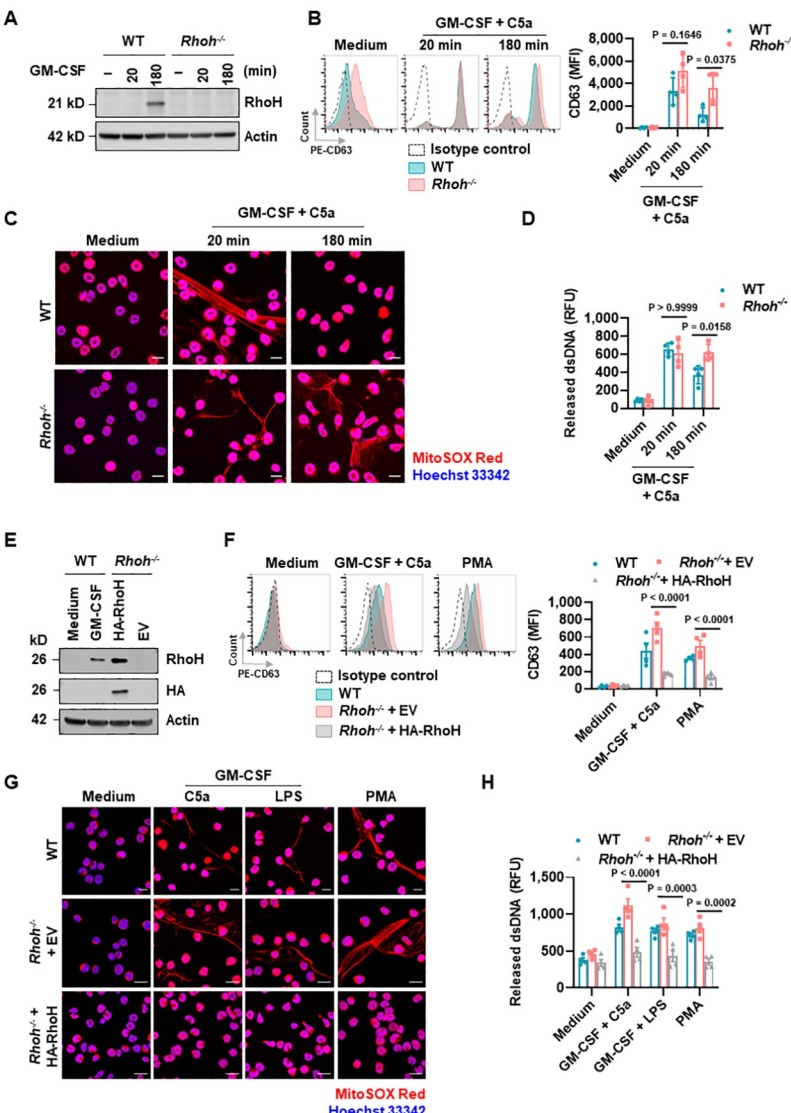

**Fig 2. Induction of RhoH expression by GM-CSF stimulation or overexpression of HA-RhoH impairs neutrophil degranulation and dsDNA release.** (**A**) Neutrophils purified from bone marrow of WT and *Rhoh⁻/⁻* mice were incubated with GM-CSF for the indicated times followed by immunoblotting. (**B–D**) Neutrophils from WT and *Rhoh⁻/⁻* were pretreated with GM-CSF for the indicated times followed by activation with C5a for 15 min. (**B**) Neutrophil degranulation was determined by flow cytometry. (**C**) Extracellular DNA fibers were analyzed by confocal microscopy. Scale bars, 10 μm. (**D**) Quantification of released dsDNA in the culture supernatants. (**E**) Expression of RhoH in the indicated HoxB8 neutrophils was analyzed by immunoblotting. (**F–H**) Mature HoxB8 neutrophils were treated with the indicated stimuli. (**F**) Neutrophil degranulation was determined by flow cytometry. (**G**) Extracellular DNA fibers were analyzed by confocal microscopy. Scale bars, 10 μm. (**H**) Quantification of released dsDNA in culture supernatants. (**A, C, E, G**) Similar results were obtained from 3 independent experiments. (**B, D, F, H**) Each symbol represents a single experiment. Values are means ± SD. (**B, D**) Two-way ANOVA with Tukey's multiple comparisons test. (**F, H**) Two-way ANOVA with Šídák's multiple comparisons test. The underlying data for Fig 2B, 2D, 2F, and 2H can be found in S1 Data. The underlying data for Fig 2A and 2E can be found in S1 Raw images. dsDNA, double-stranded DNA; GM-CSF, granulocyte/macrophage colony-stimulating factor; WT, wild-type.

Other neutrophil agonists known to induce NET formation [35,36], such as PMA, showed similar results (S3C and S3D Fig).

We next generated stem cell factor (SCF)-dependent, conditional HoxB8-immortalized (henceforth HoxB8) myeloid progenitor cells from WT and *Rhoh⁻/⁻* mice. Neutrophils

differentiated from these cells have been validated as a useful model for functional studies [35,36]. To determine whether the defect in NET formation was directly related to RhoH, we transduced HoxB8 *Rhoh*$^{-/-}$ cells with lentiviral constructs carrying triple *hemagglutinin*-tagged *Rhoh* (Gene ID: 74734) (*HA-Rhoh*) or the corresponding empty vector (EV). After 5 days of differentiation, the re-expressed RhoH in HoxB8 *Rhoh*$^{-/-}$ neutrophils exhibited the typical segmented nucleus (S4A Fig) and increased surface expression of neutrophil marker Ly6G (S4B Fig). In addition, the stable expression of HA-RhoH was confirmed by immunoblotting (Fig 2E). The subsequent functional assays revealed that reconstitution of RhoH protein in *Rhoh*$^{-/-}$ neutrophils markedly reduced degranulation as compared to HoxB8 EV and non-transduced WT neutrophils (Fig 2F). Moreover, decreased amounts of released dsDNA and extracellular DNA fibers were also observed in HoxB8 RhoH reconstituted neutrophils (Fig 2G and 2H). Taken together, in both human and mouse neutrophils, elevated RHOH expression reduced neutrophil effector functions.

## RhoH and NMHC IIA function in activated mouse and human neutrophils

To dissect the underlying molecular mechanism involved in the inhibitory effects of RhoH on neutrophil effector functions, we sought to identify RhoH-interacting proteins in mouse neutrophils by performing HA-RhoH pulldown assay. As shown by Coomassie blue staining, a candidate protein approximately 230 kilo Daltons (kD) in size selectively bound to HA-RhoH upon GM-CSF priming and C5a activation, while this association was weak in activated EV or unstimulated HA-RhoH neutrophils (Fig 3A, top). Mass spectrometry analysis revealed this protein as non-muscle myosin heavy chain IIA (NMHC IIA) (Accession: NP_071855), which is also known as Myh9 or myosin-9, with 45.1% coverage (Fig 3A, bottom). We further verified the increased interaction between RhoH and NMHC IIA in activated mouse neutrophils (Fig 3B). Moreover, this association was also detected in human neutrophils after induction of RHOH by GM-CSF stimulation (Fig 3C).

Among 3 isoforms of non-muscle myosin II heavy chain (NMHC), termed IIA, B, and C, neutrophils express only NMHC IIA [41]. NMHC IIA functions in the form of myosin IIA and is tightly regulated at the level of folding, myosin filament assembly, actin binding, ATPase, and motor activity, as well as by interactions with other proteins [42]. The fundamental role of myosin IIA in neutrophil migration has been extensively studied in vitro and in vivo [43], yet the involvement of myosin IIA in neutrophil effector function is not fully understood. To examine the potential role of myosin IIA in neutrophil functions, we pretreated human neutrophils with either ML-7, which selectively blocks myosin light chain kinase (MLCK)-mediated activation of myosin II, or (S)-4′-nitro-blebbistatin (nBleb), a photostable and non-fluorescent blebbistatin derivative which inhibits the ATPase activity and sequesters myosin II in a low-affinity state for actin [44,45]. Upon GM-CSF priming and subsequent C5a activation, both inhibitors decreased, in a concentration-dependent manner, neutrophil degranulation (Fig 3D) and dsDNA release (Fig 3E), and, as a consequence, severely impaired the formation of extracellular traps (Fig 3F).

In addition, we reduced NMHC IIA expression with 2 different shRNA targeting *Myh9* (Gene ID: 17886) in HoxB8 cells. After differentiation into mature mouse neutrophils (S4A and S4B Fig), successful reduction of NMHC IIA expression was confirmed by immunoblotting (Fig 3G and S5A Fig). NMHC IIA-deficient cells showed defects in NET formation as evidenced by a reduction of degranulation (Fig 3H and S5B Fig) and dsDNA release (Fig 3I and 3J, and S5C and S5D Fig).

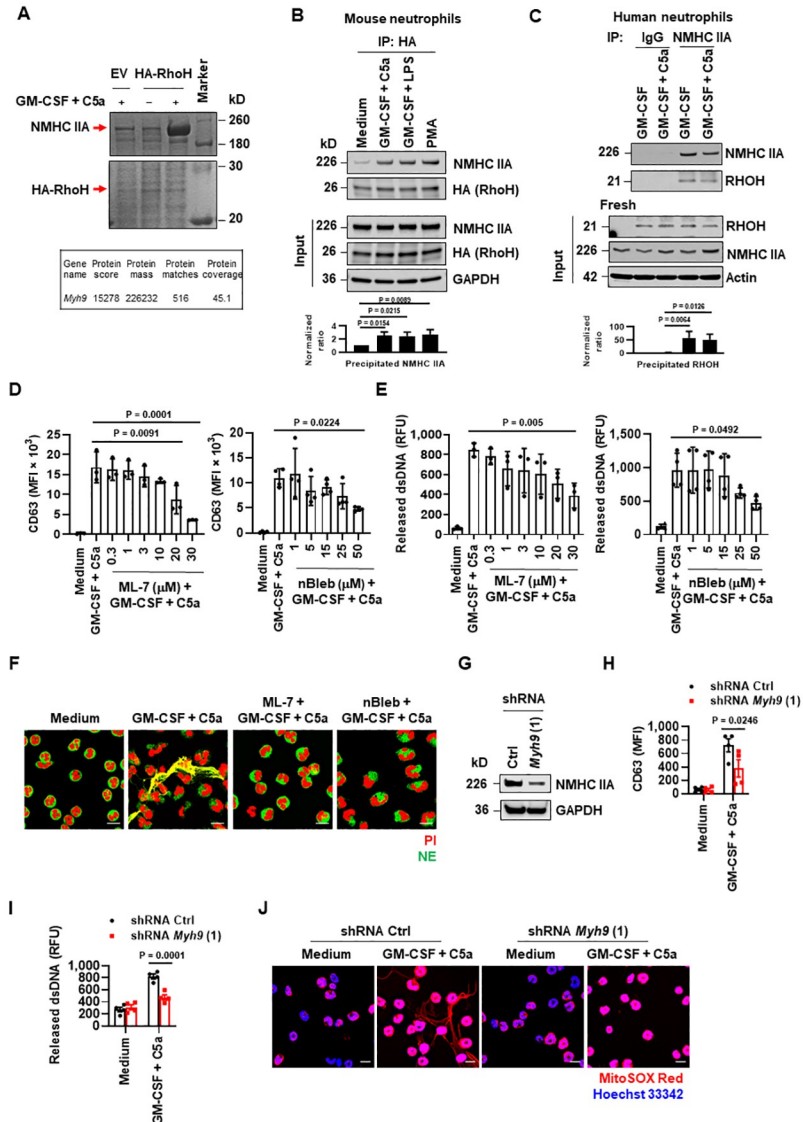

**Fig 3. RhoH interacts with NMHC IIA. (A)** Coomassie blue staining of HA pulldown proteins in HoxB8 neutrophils after stimulation with the indicated triggers. The proteins interacting with RhoH were identified by mass spectrometry analysis. **(B, C)** Immunoblot analysis after Co-IP with anti-HA antibody in Hoxb8 neutrophils **(B)** and anti-NMHC IIA antibody in human neutrophils **(C)**. Quantification of the precipitated NMHC IIA **(B)** or RHOH **(C)** was shown below the representative images. **(D–F)** Pretreated human neutrophils with vehicle control or myosin IIA inhibitors for 30 min were subsequently activated with GM-CSF in combination with C5a. **(D)** Neutrophil degranulation was determined by flow cytometry. **(E)** Quantification of dsDNA released into the supernatants. **(F)** NET formation indicated by the colocalization of NE (green) with released DNA (PI, red) was analyzed by confocal microscopy. Scale bars, 10 μm. **(G)** Protein expression of NMHC IIA in mature HoxB8 neutrophils treated with control (Ctrl) or Myh9 shRNA was analyzed by immunoblot. **(H, I)** Mature HoxB8 neutrophils were treated as indicated. **(H)** Neutrophil degranulation was determined by flow cytometry. **(I)** Quantification of released dsDNA in culture supernatants. **(J)** Extracellular DNA fibers were analyzed by confocal microscopy. Scale bars, 10 μm. **(A–C, F, G, J)** Three independent experiments were performed. **(B–E, H, I)** Values represent means ± SD. **(B–E)** One-way ANOVA with Dunnett's multiple comparisons test. **(H, I)** Two-way ANOVA with Šídák's multiple comparisons test. The underlying numerical data for Fig 3B–E, 3H, and 3I can be found in S1 Data. The uncropped immunoblots for Fig 3A–C and 3G can be found in S1 Raw images. Co-IP, co-immunoprecipitation; dsDNA, double-stranded DNA; GM-CSF, granulocyte/ macrophage colony-stimulating factor; NE, neutrophil elastase; NET, neutrophil extracellular trap; NMHC IIA, non-muscle myosin heavy chain IIA; PI, propidium iodide.

## Myosin IIA associates with neutrophil organelles to mediate their binding with actin filaments

Given the regulatory role of myosin IIA in actin reorganization, we evaluated the potential inhibition of actin dynamics by myosin IIA inhibitors. F-actin was in a ring-like manner close to the cell membrane in resting human neutrophils but accumulated asymmetrically in the cytosol after neutrophil activation (S6A Fig). ML-7 treatment caused no significant changes in the morphology of resting neutrophils but greatly reduced neutrophil spreading and F-actin formation upon activation (S6A Fig). Consistent with the previous report [43], nBleb treatment caused an increased cell area in resting neutrophils. Subsequent activation of these cells by GM-CSF plus C5a did not induce significant cell spreading additionally (S6A Fig). Short-term application of nBleb did not affect actin polymerization upon neutrophil activation (S6A Fig). In contrast to ML-7 that reduced activation of regulatory light chain (RLC), nBleb had no effects on the phosphorylation of RLC (S6B Fig). In addition, neither ML-7 nor nBleb treatment interfered with ROS production (S6C Fig).

In natural killer (NK) cells, myosin IIA is associated with lytic granules to facilitate the interaction of granules with F-actin at the immunologic synapse for exocytosis [46,47]. Mechanistically, the phosphorylation of NMHC IIA at serine 1943 (S1943) in the tailpiece enables its association with granules and act as a single molecule actin motor [48]. To ascertain if this also applied to neutrophils, we first examined the phosphorylation of NMHC IIA at S1943 by immunoblotting. Interestingly, comparable level of phosphorylation was detected in resting and activated human neutrophils (Fig 4A). We further isolated granules from resting and activated human neutrophils using density gradient ultracentrifugation and equally collected into 10 fractions and detected no traces of actin, indicating the purity of granule fractions (Fig 4B). Granule proteins were mainly detected in fractions 7 to 10 (Fig 4B), as verified by elastase activity (Fig 4C) and capability to kill *E. coli*-GFP (Fig 4D). NMHC IIA was identified in the postnuclear lysate (PNL) and partial co-fractionation with granule protein MPO, lactoferrin, and MMP-9 (Fig 4B). Consistent with the similar amount of phosphorylation at S1943 in neutrophils before and after activation (Fig 4A), the presence of NMHC IIA in granules also did not change upon neutrophil activation (Fig 4B).

Beyond granules, NMHC IIA has also been shown to interact with mitochondria membranes independently of F-actin as demonstrated previously in platelets [49]. Therefore, resting and activated human neutrophils were subjected to cell fractionation, and the mitochondrial fraction was separated from the cytosolic fraction and analyzed by immunoblotting. We detected NMHC IIA mainly in the cytosol and only traces in the isolated mitochondrial fraction, together with other known mitochondrion-localizing proteins, such as mitofusin-2 and TOMM20 (Fig 4E). We compared the distribution of NMHC IIA in the mitochondrial fraction between resting and activated human neutrophils and failed to observe any significant differences (Fig 4E). Moreover, the Pearson's coefficient value between the colocalized volume of NMHC IIA and CD63 or MTO indicated similar colocalization in resting and activated neutrophils (Fig 4F). Taken together, these results pointed to an interaction between myosin IIA and neutrophil organelles that is constantly present in resting and activated human neutrophils.

To test whether NMHC IIA enabled the loading of the associated organelles to F-actin for intracellular trafficking, we performed an actin-binding in vitro assay as previously described [47]. Isolated mouse neutrophil granules and mitochondria were incubated with exogenous F-actin and then subjected to centrifugation at a speed which is sufficient to pellet organelles, but not F-actin alone. The actin crosslinking protein, α-actinin, was used as a positive control to aggregate F-actin (Fig 4G, upper panel). As expected, the addition of granules

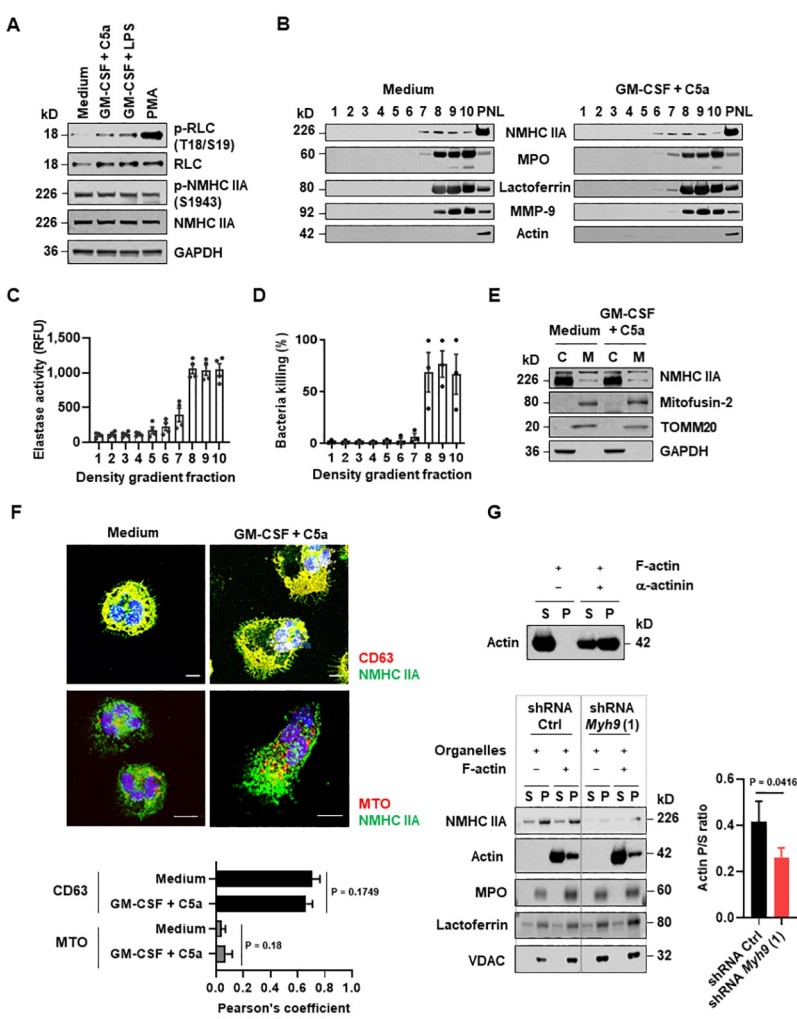

**Fig 4. NMHC IIA associates with neutrophil organelles and mediates their binding to actin filaments. (A)** Phosphorylation of myosin IIA in human neutrophils treated as indicated was analyzed by immunoblotting. Threonine 18 (T18); Serine 19 (S19); Serine 1943 (S1943). (**B–D**) The PNL from unstimulated or GM-CSF primed and C5a-activated human neutrophils was isolated and collected equally into 1–10 fractions, from top to bottom. (**B**) Immunoblot analysis of density gradient fractions. (**C**) NE activity of density gradient fractions. (**D**) Bacteria killing mediated by density gradient fractions. (**E**) Immunoblot analysis of NMHC IIA in cytosolic and mitochondrial fractions from control and activated human neutrophils. C, cytosolic fraction. M, mitochondrial fraction. (**F**) Granular or mitochondrial localization of NMHC IIA in unstimulated and activated human neutrophils was analyzed by confocal microscopy. Scale bars, 10 μm. Numerical analysis was performed on 10 cells in each group and the Pearson's coefficient in colocalized volume between NMHC IIA and CD63 or MTO was calculated using Imaris software. (**G**) Enriched granules and mitochondria from mature HoxB8 neutrophils treated with control (Ctrl) or *Myh9 shRNA* were incubated for 30 min in the presence or absence of F-actin. The pellet (P) and supernatant (S) were evaluated by immunoblotting (left). The ratios of F-actin in the pellet (P) to the supernatant (S) were quantified (right). (**A, B, E–G**) Data are representative of 3 independent experiments. (**C, D, F, G**) Values are means ± SD. (**F, G**) Unpaired 2-tailed Student *t* test was applied. The underlying numerical data for Fig 4C, 4D, 4F, and 4G can be found in S1 Data. The uncropped immunoblots for Fig 4A, 4B, 4E, and 4G can be found in S1 Raw images. GM-CSF, granulocyte/ macrophage colony-stimulating factor; NE, neutrophil elastase; NMHC IIA, non-muscle myosin heavy chain IIA; PNL, postnuclear lysate.

and mitochondria co-sedimented the F-actin in the pellet and reduced the free F-actin in the supernatant of HoxB8 neutrophils treated with control (Ctrl) shRNA (Fig 4G, lower panel). The amount of NMHC IIA associated with the granules and mitochondria decreased and therefore the percentage of sedimented F-actin in the pellet was significantly reduced in *Myh9*

shRNA compared to Ctrl treated mouse neutrophils (Fig 4G, lower panel). These findings, together with results obtained from NET formation assays (Fig 3H–3J), suggested that NMHC IIA enables the binding of granules and mitochondria to F-actin for intracellular organelle trafficking.

## RhoH inhibits myosin IIA-mediated granules and mitochondria binding to F-actin

To understand the cellular alterations underlying the interaction between RhoH and NMHC IIA upon neutrophil activation, we next evaluated the intracellular localization of NMHC IIA. Similar amounts of NMHC IIA were detected in granule and mitochondrial fractions from EV and HA-RhoH expressing neutrophils (S7A and S7B Fig). Intriguingly, RhoH was also found to localize at granules and mitochondria (S7A and S7B Fig), presumably owing to its interaction with NMHC IIA. Moreover, overexpression of RhoH reduced the ability of neutrophil granules and mitochondria to interact with F-actin (Fig 5A), corresponding with the impaired degranulation and dsDNA release seen in these cells (Fig 2F–2H).

Molecular docking analysis suggests that RHOH spatially interacted with NMHC IIA by forming hydrogen bonds with several amino acid residues (Fig 5B). Among which, alanine (Ala) 655 and Ala 659 were present at the actin-binding motifs (residues 654–676) of NMHC IIA [50]. RHOH residue lysine (Lys) 34 (K34), which belongs to the switch I domain [51], was located at the interface with NMHC IIA (Fig 5B). Sequence alignment of RHOH and the classical RHO GTPases RAC1, CDC42, RHOA indicated that K34 (depicted in red) is a RHOH-specific amino acid, which might reflect its indispensability for RhoH function (Fig 5B). We therefore proposed that RHOH may engage the actin-binding interface in NMHC IIA to impair NMHC IIA-actin interaction and induce molecular motor dysfunction for neutrophil organelle transport.

To this end, we designed specific point mutations in mouse RhoH (Accession: NP_001074574.1) and transduced HoxB8 cells with the corresponding lentiviral constructs. Upon differentiation into mature neutrophils, a point mutation of Lys 34 to Ala (K34A) in RhoH (RhoH$^{K34A}$) displayed decreased ability to interact with NMHC IIA (Fig 5C), allowing the binding of granules and mitochondria to F-actin in an actin-binding in vitro assay (Fig 5D). These cells also elicited efficient degranulation and dsDNA release following activation, in contrast to WT RhoH overexpressing HoxB8 neutrophils (Fig 5E–5G). According to the molecular docking results, the neighboring residue tyrosine (Tyr) 33 in RhoH was not involved in binding to NMHC IIA and therefore used as a mutant control (Fig 5B). Unexpectedly, the Tyr 33 to phenylalanine (Phe) (RhoH$^{Y33F}$) mutant and the combined double mutant (RhoH$^{Y33F \text{ and } K34A}$) failed to stably express at protein level despite profound mRNA expression (Fig 5C and S8A Fig). Lower expression of RhoH$^{Y33F}$ or RhoH$^{Y33F \text{ and } K34A}$ mutants appeared to be due to lysosomal degradation since the lysosomal proton pump inhibitor bafilomycin A1 (Baf A1) could block RhoH degradation (S8B Fig), as it has been previously reported in T cells [22]. Collectively, these results demonstrated that RhoH inhibits myosin IIA-mediated granules and mitochondria transport by blocking their binding to F-actin.

## RhoH impairs F-actin formation presumably by modulating Rac1 activity

Given that RHO GTPases are key regulators of the cell cytoskeleton, we further evaluated the effects of RhoH on F-actin and microtubule cytoskeleton before and after neutrophil activation. We did not observe significant difference in microtubule network between EV and HA-RhoH expressing neutrophils (S9A Fig). However, overexpression of RhoH reduced F-actin formation upon neutrophil activation as compared to EV (Fig 6A), without interfering

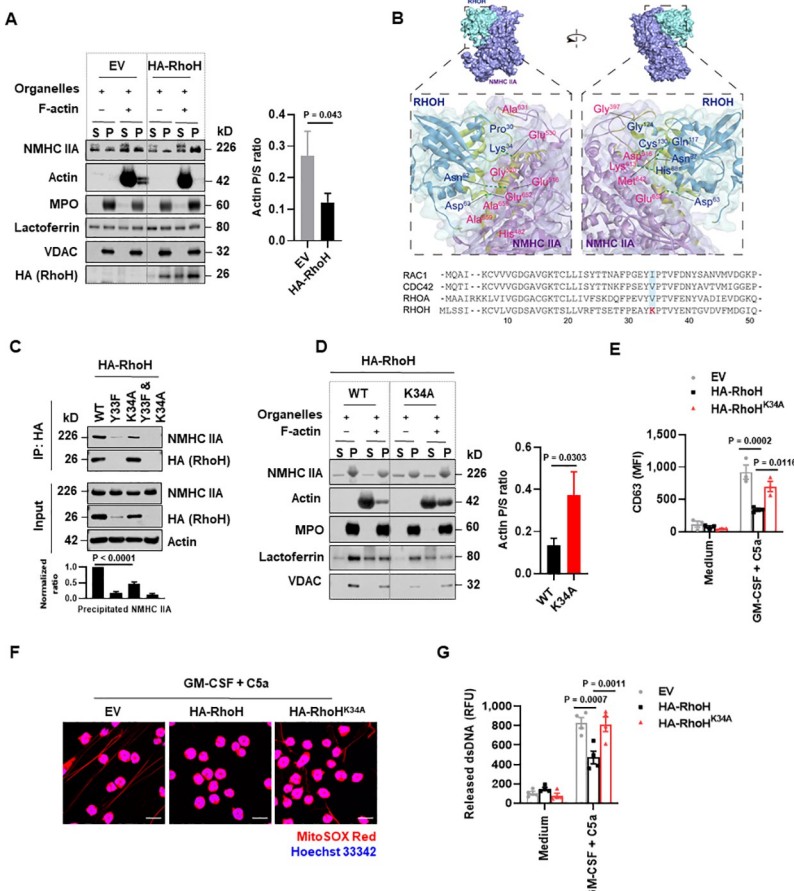

**Fig 5. RhoH inhibits NMHC IIA-mediated granules and mitochondria binding to F-actin.** (**A**) Enriched granules and mitochondria from mature HoxB8 neutrophils expressing HA-RhoH or the corresponding EV were incubated for 30 min in the presence or absence of F-actin. After centrifugation, the pellet (P) and supernatant (S) were evaluated by immunoblotting (left). The ratios of F-actin in the pellet (P) to the supernatant (S) were quantified (right). (**B**) Docking analysis for predicting the binding sites between RHOH and NMHC IIA (upper). Sequence alignment of RAC1, CDC42, RHOA, and RHOH (lower). (**C**) Immunoblot analysis of Co-IP with anti-HA antibody in cell lysates from activated HoxB8 neutrophils expressing WT or mutated HA-RhoH. Quantification of the precipitated NMHC IIA was shown below the representative images. (**D**) Enriched granules and mitochondria were incubated for 30 min in the presence or absence of F-actin. The pellet (P) and supernatant (S) were evaluated by immunoblotting (left). The ratios of F-actin in the pellet (P) to the supernatant (S) were quantified (right). (**E–G**) Mature HoxB8 neutrophils expressing WT or mutated HA-RhoH or EV were treated as indicated. (**E**) Neutrophil degranulation was determined by flow cytometry. (**F**) Extracellular DNA fibers were analyzed by confocal microscopy. Scale bars, 10 μm. (**G**) Quantification of dsDNA released into the supernatants. (**A, C, D, F**) Data are representative of 3 independent experiments. Values are means ± SD. Two-tailed Student $t$ test (**A, D**); 1-way ANOVA with Dunnett's multiple comparisons test (**C**); 2-way ANOVA with Tukey's multiple comparisons test (**E, G**). The underlying numerical data for Fig 5A, 5C–E, and 5G can be found in S1 Data. The uncropped immunoblots for Fig 5A, 5C, and 5D can be found in S1 Raw images. Co-IP, co-immunoprecipitation; dsDNA, double-stranded DNA; EV, empty vector; NMHC IIA, non-muscle myosin heavy chain IIA; WT, wild-type.

with ROS production (S9B Fig), which plays an important role in regulating F-actin polymerization. Defective F-actin formation was reversed by K34A mutation in RhoH (Fig 6A). Moreover, RhoH had no effects on the phosphorylation of RLC and NMHC IIA of myosin IIA upon neutrophil activation (Fig 6B), indicating that other factors are involved in the regulation of F-actin formation by RhoH.

RHOH has previously been reported to modulate the activities of other RHO GTPases such as RAC, CDC42, and RHOA [52,53], therefore, we next determined the activation of these

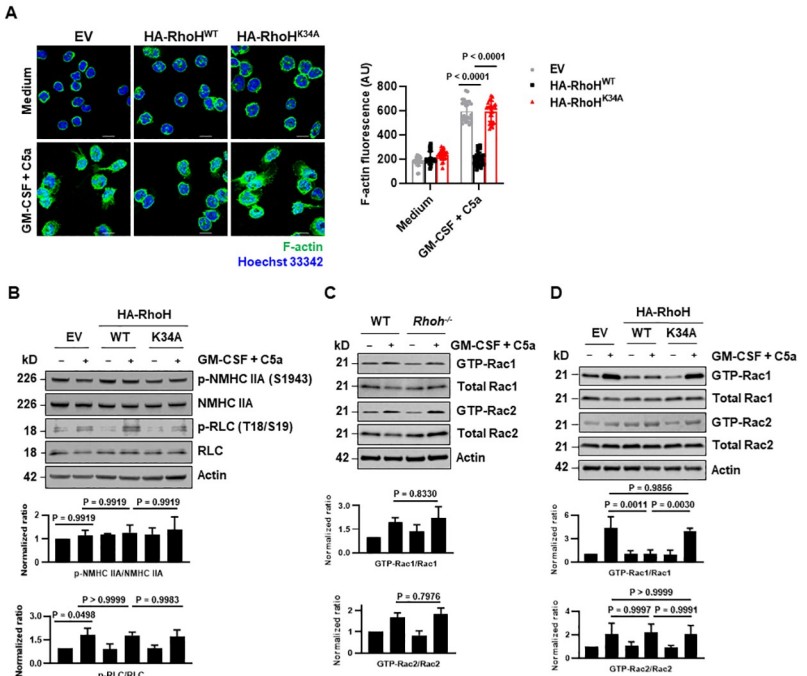

**Fig 6. RhoH impairs actin rearrangement presumably by modulating Rac1 activity.** (**A, B**) Mature HoxB8 neutrophils expressing WT or mutated HA-RhoH or EV were treated as indicated. (**A**) F-actin distribution was analyzed by confocal microscopy (left). Scale bars, 10 μm. Quantification of F-actin fluorescence intensity was performed by automated analysis of microscopic images using Imaris software. The 25 images (each containing 8–12 cells) from 3 independent experiments were included for each condition (right). (**B**) Phosphorylation of myosin IIA was analyzed by immunoblotting. (**C, D**) The levels of GTP-bound and total Rac1 and Rac2 proteins in the indicated mouse bone marrow (BM) neutrophils (C) and HoxB8 neutrophils (D) were examined by immunoblotting following pulldown assay. All data are representative of 3 independent experiments. Values are means ± SD. One-way ANOVA with Tukey's multiple comparisons test was applied. The underlying numerical data for Fig 6A–D can be found in S1 Data. The uncropped immunoblots for Fig 6B–D can be found in S1 Raw images. EV, empty vector; WT, wild-type.

RHO GTPases by effector pulldown assays. Similar amounts of GTP-bound Cdc42 (S9C Fig) and GTP-bound RhoA (S9D Fig) were observed upon stimulation of neutrophils expressing HA-RhoH and EV. *Rhoh* deficiency did not cause significant changes in Rac1 and Rac2 activation (Fig 6C). RhoH overexpression reduced Rac1 activity, which was abolished by K34A mutation in RhoH (Fig 6D). The inhibitory effect was not observed on Rac2 activation (Fig 6D). Thus, RhoH inhibits F-actin formation presumably by modulating Rac1 activity.

## *Rhoh*$^{-/-}$ neutrophils exhibit augmented NET formation and bacteria killing in a mouse model of peritonitis

In order to verify the regulatory role of RhoH in vivo, we used an acute murine peritonitis model in which a low intraperitoneal dose of GFP-*E. coli* is applied [54]. Consistent with our results obtained in in vitro stimulation experiments (Fig 2A), RhoH was up-regulated in peritoneal cells in a time-dependent manner, with abundant expression observed at 8 h following *E. coli* challenge (Fig 7A). Compared to WT, bacterial loads at 8 h after infection were significantly reduced in *Rhoh*$^{-/-}$ mice (Fig 7B). NETs were likely to be involved in the observed bacteria killing, as large amount of dsDNA (Fig 7C) and NE (Fig 7D) were detected in peritoneal lavage fluid (PLF) supernatants. In addition, higher levels of dsDNA and NE were detected in PLF obtained from *Rhoh*$^{-/-}$ mice (Fig 7C and 7D).

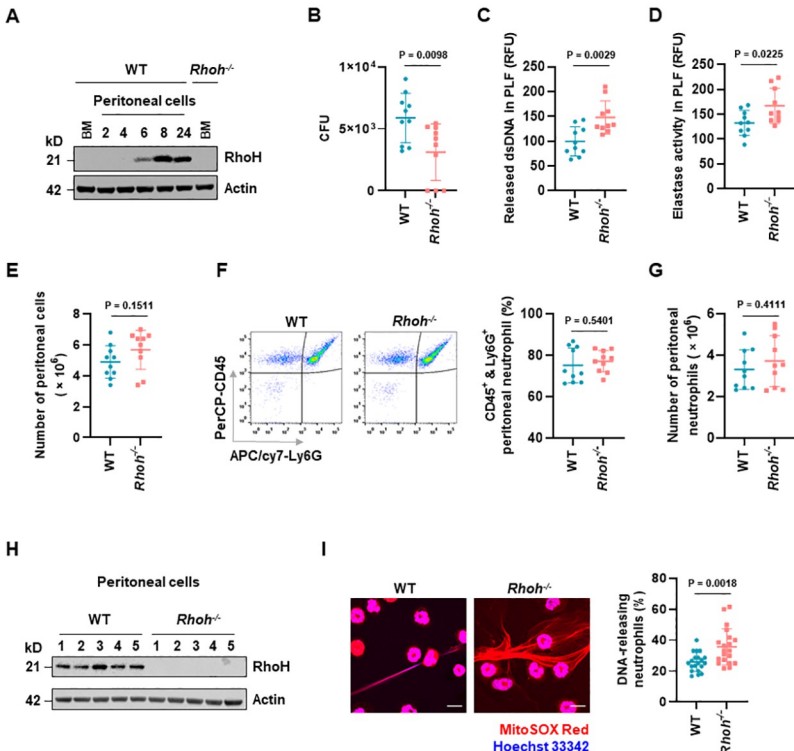

**Fig 7.** *Rhoh*^(-/-) **neutrophils display enhanced NET formation and bacteria killing in experimental peritonitis.** (**A**) Peritoneal cells from WT mice challenged intraperitoneally with *E. coli* was analyzed by immunoblot. Protein lysates of bone marrow (BM) neutrophils from WT and *Rhoh*^(-/-) mice were loaded as negative control. Data were representative of 3 independent experiments. (**B–I**) WT and *Rhoh*^(-/-) mice were challenged intraperitoneally with *E. coli* and sacrificed after 8 h; *n* = 10 for each group unless mentioned otherwise. (**B**) Bacteria burden was quantified as CFU. (**C**) Quantification of released dsDNA in PLF supernatants. (**D**) Quantification of NE activity in PLF supernatants. (**E**) Total peritoneal cell count. (**F**) Infiltrated peritoneal neutrophils were defined as CD45^+ and Ly6G^+. Left, representative original flow cytometry data. Right, proportion of peritoneal neutrophils. (**G**) Number of peritoneal neutrophils. (**H**) Protein level of RhoH in peritoneal cells was analyzed by immunoblot (*n* = 5). (**I**) Representative confocal images of peritoneal cells stained with MitoSOX Red and Hoechst 33342, left. Scale bars, 10 μm. Quantification of cells releasing dsDNA fibers was performed in 2 randomly selected images from each mouse, right. Values are means ± SD. Statistical analyses were performed using unpaired 2-tailed Student *t* test. The underlying data for Fig 7B–E, 7G, and 7I can be found in S1 Data. The underlying data for Fig 7A and 7H can be found in S1 Raw images. CFU, colony-forming unit; dsDNA, double-stranded DNA; NE, neutrophil elastase; NET, neutrophil extracellular trap; PLF, peritoneal lavage fluid; WT, wild-type.

Enhanced bacteria killing and increased accumulation of NETs components may result from more neutrophils recruited to the inflamed peritoneal cavities of *Rhoh*^(-/-) mice. However, although *E. coli* injection resulted in abundant accumulation of immune cells (Fig 7E) and neutrophils represented the dominant cell type (Fig 7F), we failed to observe significant differences in the absolute number of neutrophils between WT and *Rhoh*^(-/-) mice (Fig 7G). Since we detected up-regulated RhoH expression in infiltrating cells in WT mice (Fig 7H), we speculated that perturbed neutrophil activity may be the reason. Indeed, *Rhoh*^(-/-) neutrophils displayed enhanced ability in forming extracellular dsDNA fibers (Fig 7I). Collectively, RhoH knockout results in enhanced NET formation and host defense against bacteria, confirming RhoH as a molecular brake on neutrophil effector functions under in vivo conditions.

## Discussion

It is becoming increasingly clear that the local microenvironment shapes neutrophil functional responses by triggering neutrophil transcription reprogramming, which represents a

mechanism driving neutrophil diversity and plasticity in diseases. Nevertheless, neutrophil intrinsic regulators involved in this process remain largely unknown. In this study, we demonstrate that neutrophils exhibit blunted functional responses after induction of RhoH by CF serum incubation, GM-CSF stimulation, or genetic manipulation in vitro. Knocking out *Rhoh* restores neutrophil functional activity as reflected by enhanced degranulation and NET formation as well as augmented host defense against *E. coli* in an experimental peritonitis mouse model. These data suggest that RHOH acts as an inducible molecular brake that limits neutrophil responses during inflammation.

A number of studies have endeavored to explain the paradox that massive neutrophils infiltrate in the lung of CF patients but fail to eradicate bacterial infections. Some reports indicate that impaired degranulation, NET formation, and microbial killing are caused by an intrinsic CFTR defect and can be corrected by CFTR potentiator therapy [55,56]. However, other reports suggest that the canonical functions of CF neutrophils are modulated by the local airway microenvironment [13]. Moreover, blocking de novo transcription in CF airway neutrophils restores their bactericidal activity [13]. In this study, we show that human neutrophils treated with serum from CF patients suffering from systemic inflammation acquire the phenotype of CF neutrophils, with up-regulated RHOH expression associated with impaired degranulation and NET formation. These data provide evidence for the notion that adaptive changes of CF neutrophils are conditioned by chronic inflammation and highlight RHOH as an intrinsic regulator of neutrophil plasticity in the course of inflammation.

GM-CSF is an important hematopoietic growth factor and immune modulator. Short-term in vitro exposure to GM-CSF, so-called priming, favors functional responses of neutrophils [30,57–59], while long-term stimulation with GM-CSF alters the transcriptional profile of neutrophils [28,38]. Of note, *RHOH* is a prominent gene up-regulated by GM-CSF [28,38]. GM-CSF has been shown to be up-regulated either systemically and/or in tissues in various diseases [60,61], implying that RHOH might represent a common transcriptional adaptation in distinct inflammatory conditions. Indeed, in addition to CF, we also provide evidence for enhanced *RHOH* expression in neutrophils from COVID-19 and cancer patients. It would be interesting to investigate whether the newly identified RHOH-mediated intracellular regulatory axis also results in decreased functional activities of neutrophils in these diseases.

The functional role of RHOH in HPCs, T cells, B cells, eosinophils, and mast cells has been well elucidated, while our knowledge of RHOH in neutrophils remains obscure. Herein, we show that RHOH interacts with NMHC IIA in a cell activation manner. In the absence of cell activation, the phosphorylation level of RLCs is low and the head domains and the tails of myosin IIA interact to keep the molecule in an inactive compact structure. Upon neutrophil activation, enhanced phosphorylation of RLCs by kinases, such as MLCK, disrupts the head–tail interactions and releases myosin IIA in an open conformation. In response to GM-CSF and/or other inflammatory factors, neutrophils increase the expression of RHOH, which in turn binds to the active myosin IIA and reduces the binding ability of NMHC IIA to F-actin. A point mutation of Lys 34 to Ala in RhoH (RhoH$^{K34A}$) decreases the interaction between RhoH and NMHC IIA and restores the NMHC IIA-mediated binding of granules and mitochondria to F-actin, hence partially releasing the inhibitory effects of RhoH on degranulation and NET formation. To our knowledge, this is the first study that demonstrates the interaction between RHOH and NMHC IIA. Recent studies have linked both RHOH and NMHC IIA to cancers [53,62,63]. Therefore, studying the potential interaction between RHOH and NMHC IIA might offer new opportunities to understand their role in cancer.

Conflicting roles of RHOH in actin dynamics have been reported. On the one hand, RhoH does not regulate actin reorganization in NIH3T3 or MDCK cells [17]. On the other hand, RHOH is involved in the regulation of the actin cytoskeleton by modulating the activity of

classical RHO proteins [23,52,64]. In the current study, we report that aberrant expression of RhoH impairs F-actin formation following neutrophil activation. Although RhoH does not affect the activation of myosin IIA, reduced RhoH-myosin IIA interaction is associated with rescued F-actin formation. We further show RhoH specifically inhibits Rac1 activity and releases the inhibition once RhoH-myosin IIA interaction is disrupted. It is likely that RhoH modulates F-actin through Rac1; however, the mechanism by which RhoH suppresses Rac1 activation remains unresolved. We cannot rule out the possibility that NMHC IIA might be involved in Rac1 activation, as shown in a previous study [65]. As guanine nucleotide exchange factors (RhoGEFs) and GTPase-activating proteins (RhoGAPs) coordinate the activation of the Rho family of GTPases, another possibility is RhoH acts on these proteins to modulate the activation of Rac1.

The actin cytoskeleton and their motors, myosin superfamilies, are parts of a well-known machinery that actively transports intracellular organelles in eukaryotic cells [66]. However, it remains unclear in neutrophils how the cytoskeleton and the related motor proteins recognize and transport their cargoes. Here, we propose a working model in which NMHC IIA links neutrophil organelles to the F-actin cables. We show a permanent phosphorylation of NMHC IIA at S1943 in neutrophils. NMHC IIA constitutively interacts with the surface of mitochondria and granules, which enables rapid assembly of the actomyosin transport machinery, thus supporting degranulation and the production of NETs within minutes after stimulation. Therefore, our data confirm the previously reported function of myosin IIA in terms of translocation of granules and mitochondria in other cell types such as NK cells and thrombocytes [46–49]. However, there might be other myosin proteins involved in the intracellular organelle translocation. For example, a recent study demonstrated that Myo19 binds and tethers damaged mitochondria to actin for removal through a process termed mitocytosis [67].

Our study was designed to evaluate the impact of aberrant RHOH expression in neutrophil effector functions, which is found in neutrophils under various inflammatory conditions, including but not limited to CF, and to delineate the underlying mechanism. Due to COVID-19 restrictions related to in-person visits for CF patients, the sample size for experiments has been limited. Instead, we mainly used freshly isolated human blood and mouse bone marrow neutrophils in combination with genetic modification for functional assays. In addition, the regulatory role of RhoH was further evaluated in an *E. coli*-induced peritonitis model. Both our in vitro and in vivo data consistently suggest that RhoH acts as an inducible intracellular brake for neutrophil effector functions. However, the molecular mechanism inducing RhoH expression remains to be identified.

## Materials and methods

### Reagents

Human GM-CSF was purchased from Novartis (Basel, Switzerland). Mouse GM-CSF was from R&D Systems (Abingdon, United Kingdom). Human and mouse C5a were supplied by Hycult Biotech (Uden, the Netherlands). Phorbol-12-myristate-13-acetate (PMA) and Triton X-100 were from Merck Millipore (Burlington, Massachusetts, United States of America); 4-hydroxytamoxifen (4-OHT), lipopolysaccharide (LPS, 055:B5), bovine serum albumin (BSA), glutaraldehyde, saponin, protease inhibitor cocktail, sodium borohydride (NaBH4), dihydrorhodamine 123 (DHR 123), cytochalasin B, ML-7, tris base, and sodium dodecyl sulfate (SDS) were from Sigma-Aldrich (Buchs, Switzerland). MitoSOX Red, Alexa Fluor 488 Phalloidin, MitoTracker Orange CM-H2 TMRos, Prolong Gold mounting medium, Hoechst 33342, the Quant-iT PicoGreen dsDNA assay kit, propidium iodide (PI), Hanks' balanced salt solution (HBSS), RPMI-1640/GlutaMAX medium, penicillin/streptomycin and Pierce BCA

protein assay kit were all obtained from Thermo Fisher Scientific, distributed by LuBioScience GmbH (Luzern, Switzerland). (S)-4′-nitro-blebbistatin was from Cayman Chemical, distributed by Adipogen AG (Liestal, Switzerland), IQ SYBR Green Supermix was from Bio-Rad Laboratories AG (Cressier, Switzerland). X-VIVO 15 medium without phenol red and antibiotics was from Lonza (Walkersville, Maryland, USA). Human IgG polyvalent was a kind gift from CSL Behring (Bern, Switzerland). Normal goat (Cat # NBP2-23475, dilution 1:400), rat (Cat # NBP2-33356, dilution 1:400), and rabbit (Cat # NBP1-71681, dilution 1:400) sera were purchased from Novus Biologicals, Abingdon, UK. ChromPure human IgG (Cat # 009-000-003, dilution 1:1,000) and normal mouse sera (Cat # 015-000-120, dilution 1:400) were obtained from Milan Analytica AG (Rheinfelden, Switzerland). DNase I was supplied by Worthington Biochemical Corporation (Lakewood, New Jersey, USA). Mouse SCF-secreting Chinese hamster ovary cells (CHO/SCF) were kindly provided by Dr. Häcker (University of Freiburg, Germany). The Mouse Neutrophil Enrichment Kit was from Stemcell Technologies (Köln, Germany). RhoA/Rac1/Cdc42 Activation Assay Combo Biochem Kit was from Abcam (Cambridge, UK). Pancoll-Human was purchased from PAN-Biotech GmbH (Aidenbach, Germany). Glass coverslips (12 mm diameter) for immunofluorescence staining (Cat # 1001/12) were purchased from Karl Hecht "Assistent" GmbH (Sondheim/Rhön, Germany). Black, glass-bottom 96-well and white 96-well plates were from Greiner Bio-One GmbH (Frickenhausen, Germany).

## Mice

*Rhoh*⁻ᐟ⁻ mice with a C57BL/6J background were generated and provided by Dr. C. Brakebusch (Department of Molecular Pathology, University of Copenhagen, Copenhagen, Denmark) [18]. WT mice on C57BL/6J background were bred germfree at the Clean Mouse Facility, University of Bern, Switzerland. Mice were housed in specific-pathogen-free (SPF) conditions and were euthanized using carbon dioxide gas before material harvest. Sex- and age-matched mice were used in our experiments. Mouse studies were approved by the Cantonal Veterinary Office of Bern, Switzerland (license number 40/09) and carried out in accordance with the Swiss federal legislation on animal welfare.

## Isolation of human blood and mouse bone marrow neutrophils

Human blood neutrophils from healthy individuals and CF patients were isolated by Ficoll-Hypaque centrifugation as described previously [35]. Written, informed consent was obtained from all blood donors and this study was approved by the Ethics Committee of Canton Bern (approval number 18/2001). The purified populations contained >95% neutrophils as assessed by staining with the Hematocolor Stain Set (Merck Millipore) followed by light microscopic analysis.

Primary mouse bone marrow neutrophils were isolated from WT and *Rhoh*⁻ᐟ⁻ mice with a negative selection technique using the EasySep Mouse Neutrophil Enrichment Kit (Stemcell Technologies) [35]. Neutrophil purity was always higher than 85% as assessed by Diff-Quik staining and light microscopy.

## Genetic modifications in SCF-HoxB8 cells

Murine *HA*-tagged *Rhoh* (*HA-Rhoh*) cDNA containing vector EX-Mm24061-Lv118 and the corresponding EV EX-NEG-Lv1118 were purchased from GeneCopoeia, (Rockville, Maryland, USA). To generate point mutations in HA-RhoH, the QuikChange II XL Site-Directed Mutagenesis Kit (Cat # 200521, Agilent Technologies, Basel, Switzerland) was used. Mutation primers were as follows: HA-RhoH_Tyr33 forward: 5′-CCT TCC CGG AGG CCT TCA AAC

CCA CGG TGT-3′; HA-RhoH_Tyr33 reverse: 5′-ACA CCG TGG GTT TGA AGG CCT CCG GGA AGG-3′; HA-RhoH_Lys34 forward: 5′-TCC CGG AGG CCT ACG CAC CCA CGG TGT ACG-3′; HA-RhoH_Lys34 reverse: 5′-CGT ACA CCG TGG GTG CGT AGG CCT CCG GGA-3′; HA-RhoH_Tyr33Lys34 forward: 5′-GAC CTT CCC GGA GGC CTT CGC ACC CAC GGT GTA-3′; and HA-RhoH_Tyr33Lys34 reverse: 5′-GTA CAC CGT GGG TGC GAA GGC CTC CGG GAA GGT C-3′. All mutations were introduced by PCR following the manufacturer's instructions. The resulting constructs were verified by sequence analysis. Sequencing primers: forward: 5′-CGG TGG GAG GTC TAT ATA AGC AG-3′, reverse: 5′-ATT GTG GAT GAA TAC TGC C-3′. Bacterial glycerol stock harboring sequence-verified shRNA of mouse *Myh9* lentiviral plasmid vector: shRNA Myh9 (1), sequence: CGG TAA ATT CAT TCG TAT CAA; shRNA *Myh9* (2), sequence: GCC ATA CAA CAA ATA CCG CTT or the control pLKO.1-puro vector were obtained from Sigma-Aldrich (Mission shRNA). Virus was produced by transfecting HEK-293T cells with the plasmid of interest using a calcium phosphate transfection method [68]. The supernatants were collected after 24 h, filtered through a 0.22 μm filter (Merck Millipore) and stored at −80°C before use.

SCF-dependent, conditional HoxB8-immortalized myeloid progenitors were generated from WT and *Rhoh⁻/⁻* mice as previously described [69]. HoxB8 cells were passaged in RPMI 1640/GlutaMAX with 10% FCS, 100 U/ml penicillin, 100 μg/ml streptomycin, 5% SCF (CHO/SCF cell conditioned medium), and 100 nM 4-OHT. To transfer the gene of interest, 1 ml of viral supernatant was added to $5 \times 10^5$ HoxB8 cells in the presence of 16 μg of Polybrene (Sigma-Aldrich) and 100 nM 4-OHT. Transduced cells were selected with antibiotic starting 48 h after transduction and transfection efficiency was evaluated by immunoblot.

To initiate differentiation into neutrophils, $3 \times 10^4$ cells/ml were washed and then cultured in the same medium in the absence of 4-OHT. On day 3, 5 μg/ml granulocyte colony-stimulating factor (G-CSF) (PeproTech EC, London, UK) was added to cell culture and after further incubation for 2 days, cells were washed with PBS and resuspended with X-VIVO 15 medium before being used for subsequent experiments. Nuclear morphology was assessed by Diff-Quik staining and light microscopy. Maturation of the differentiated neutrophils was confirmed by measuring the cell surface expression of Ly6G using flow cytometry (FACSVerse, BD Biosciences, San Jose, California, USA).

### Neutrophil treatment

Human and mouse neutrophils as well as the differentiated HoxB8 neutrophils were cultured in X-VIVO 15 medium, primed with 25 ng/ml GM-CSF for 20 min, and subsequently stimulated with $10^{-8}$ M C5a or 100 ng/ml LPS or alternatively with 25 nM PMA for 15 min in the absence of priming. In certain experiments, human neutrophils were incubated with 5% serum from HDs or CF patients for 6 h or pretreated with inhibitors for 30 min. Primary mouse neutrophils were pretreated with GM-CSF for 3 h, before being activated as mentioned above.

### Quantification of released dsDNA in culture supernatants

Neutrophils ($2 \times 10^6$ in 500 μl X-VIVO 15 medium) were stimulated as mentioned above. At the end of the activation, a low concentration of DNase I (2.5 U/ml) was added for an additional 10 min. Reactions were stopped by addition of 2.5 mM EDTA (pH 8.0). Cells were centrifuged at 1,400 rpm for 5 min. A total of 100 μl supernatants were used to incubate with PicoGreen dye in black, glass-bottom 96-well plates for 5 min at room temperature and the fluorescence was measured with a spectrofluorometer (SpectraMax M2, Molecular Devices, San Jose, California, USA).

## Degranulation assay

Degranulation of human and mouse neutrophils was determined by measuring the increase in plasma membrane expression of surrogate markers using the following monoclonal antibodies: APC-conjugated Mouse anti-Human CD63 (clone H5C6; BioLegend, San Diego, USA), FITC-conjugated Mouse anti-Human CD66b (clone G10F5; BioLegend), and PE-conjugated Rat anti-Mouse CD63 (clone NVG-2; BD Biosciences). Isotype controls were used at the same concentration as the above antibodies: APC-conjugated Mouse IgG1, κ isotype control (clone MOPC-21; BioLegend), FITC-conjugated Mouse IgM, κ isotype control (clone MM-30; Bio-Legend), and PE-conjugated Rat IgG2a, κ isotype control (clone R35-95; BD Biosciences). Briefly, neutrophils ($5 \times 10^5$ in 200 µl X-VIVO 15 medium) were stimulated as mentioned above. In the final 5 min of priming, cytochalasin B (5 µm) was added to the cell suspension. The reaction was stopped by cold PBS. After centrifugation at 1,400 rpm for 5 min, cells were incubated with blocking buffer (for human cells: 10% FCS with 10% human IgG polyvalent in PBS; for mouse neutrophils: 2% FCS, 3% NRS, 5% hamster serum, 20% 2.4G2 supernatant in PBS) on ice for 10 min prior to incubation with antibodies for 30 min in dark. After washing with PBS supplemented with 0.4% BSA + 1 mM EDTA, cells were fixed and analyzed by flow cytometry (FACSVerse, BD Biosciences) followed by quantification using FlowJo software (Tree Star, Ashland, Oregon, USA).

## Measurement of ROS

Neutrophils ($4 \times 10^5$ in 100 µl X-VIVO 15 medium) were stimulated as mentioned above. Approximately 1 µm DHR 123 was added to the cells after priming. The reaction was stopped by adding 200 µl of cold PBS, and the ROS activity was immediately measured by flow cytometry (FACSVerse, BD Biosciences) and quantified using FlowJo software (Tree Star, Ashland, Oregon, USA).

## Immunofluorescence

Neutrophils were seeded on 12-mm glass coverslips in a 24-well plate and treated as mentioned above. Subsequently, cells were fixed with 4% paraformaldehyde for 10 min, washed twice with PBS, and permeabilized using 0.05% saponin in PBS for 3 min at room temperature. Nonspecific binding was prevented by preincubation of the cells with a blocking buffer (7.5% BSA, 30% human IgG polyvalent, 30% normal goat serum, and 1% ChromPure human IgG in PBS) at room temperature for 30 min.

To visualize extracellular traps, neutrophils were incubated with monoclonal mouse anti-human NE antibody (Dako, Baar, Switzerland) diluted in blocking solution at 4˚C overnight. After incubation with primary antibody, cells were washed with PBS and incubated with Alexa-488 goat anti-mouse secondary antibody (Thermo Fisher Scientific) at room temperature for 1 h protected from the light. DNA was stained with PI (10 µg/ml). In addition, staining with MitoSOX Red (5 µm) was performed in live cells prior to fixation. Nuclei were stained with 1 µg/ml Hoechst 33342 for 10 min at room temperature protected from the light.

To monitor actin reorganization, F-actin was stained with 1.25% Alexa Fluor 488 phalloidin in PBS plus 0.05% BSA and 0.05% of saponin for 15 min at room temperature, protected from light. Staining with MitoTracker Orange (1 µm) was performed in live cells prior to fixation. Cells were washed 3 times in PBS and counterstained with 1 µg/ml Hoechst 33342 to visualize the nuclei.

For microtubule reorganization assessment, neutrophils were fixed in mixture of 4% paraformaldehyde and 0.1% glutaraldehyde in PBS and then permeabilized with 0.5% Triton X-100 for 15 min, followed by incubation with 0.5% SDS for 15 min at room temperature.

Autofluorescence was quenched with 0.5 mg/ml NaBH4 in PBS for 10 min on ice. After blocking, samples were incubated with anti-α-tubulin monoclonal antibody (Sigma-Aldrich) diluted in blocking buffer overnight at 4°C. Samples were subsequently washed with PBS and further incubated with the secondary antibody, Alexa-555 goat anti-mouse IgG (Thermo Fisher Scientific) for 1 h at room temperature. Following extensive washing with PBS, samples were counterstained with 1 μg/ml Hoechst 33342 for 10 min in the dark.

For those experiments in which granules or mitochondria were co-stained with NMHC IIA, cells were firstly labeled with APC-conjugated anti-human CD63 or MitoTracker Orange as described above. Subsequently, cells were incubated with the primary anti-NMHC IIA antibody (1:200, Abcam) and the secondary goat anti-rabbit 532 (1:200, Dako) antibody. Nuclei were stained with 1 μg/ml Hoechst 33342 for 10 min in the dark.

Slides were mounted in Prolong Gold mounting medium and images were obtained using confocal laser scanning microscopy LSM 700 (Carl Zeiss Micro Imaging, Jena, Germany) at a resolution of $1,024 \times 1,024$ with a $\times 63/1.40$ oil DIC objective and analyzed with Imaris software (Bitplane AG, Zürich, Switzerland).

## Quantitative real-time PCR

Total RNA was extracted from neutrophils using the Quick-RNA MicroPrep Kit (Zymo Research, Irvine, California, USA) and reverse transcribed to cDNA with SuperScript III Reverse Transcriptase Kit (Thermo Fisher Scientific). Quantitative PCR was carried out with the iTaq Universal SYBR Green Supermix (Bio-Rad Laboratories, Cressier, Switzerland) using the CFX Connect Real-Time PCR Detection system (Bio-Rad Laboratories). The following primers were used [27]: mouse *Rhoh* forward: 5′-GAC CTT CCC GGA GGC CTA CA-3′ and mouse *Rhoh* reverse: 5′-TGC CGG CAG TGT CCC AGA GA-3′; mouse *18S* forward: 5′-ATC CCT GAG AAG TTC AG CA-3′ and mouse *18S* reverse:5′-CCT GGG TCA TCA TCA GGC TT-3′. Primers were synthesized by Microsynth AG, Balgach, Switzerland.

## Immunoblotting

Neutrophils were washed twice with cold PBS and lysed in lysis buffer containing 50 mM Tris (pH 7.4), 150 mM NaCl, 10% glycerol, 1% Triton X-100, 2 mM EDTA, 10 mM sodium pyrophosphate, 50 mM sodium fluoride, and 200 μm sodium orthovanadate. Shortly before use, 100 μm PMSF, a protease inhibitor cocktail and phosphatase inhibitors (Sigma-Aldrich) were freshly added into the lysis buffer. After incubation in lysis buffer on ice for 30 min, cell lysates were centrifuged at 13,000 rpm for 10 min at 4°C and the supernatant was collected. Thereafter, the protein concentration was measured with the BCA Protein Assay Kit (Thermo Fisher Scientific). Samples were heated at 95°C for 5 min. Proteins were separated by SDS-PAGE and transferred onto PVDF membranes (Immobilion-P; Merck Millipore). Membranes were incubated with a blocking buffer consisting of Tris-buffered saline (pH 7.4) with 0.1% Tween 20 (TBST) and 5% non-fat dry milk at room temperature for 1 h and then incubated with primary antibodies at 4°C overnight. The primary antibodies used as follows: anti-RhoH [26,27], anti-HA (1:1,000, Roche Diagnostics, Rotkreuz, Switzerland); anti-p-RLC (T18/S19) (1:500, Cell Signaling Technology, Danvers, Massachusetts, USA); anti-RLC (1:1,000, Cell Signaling Technology); anti-NMHC IIA (1:2,000, Abcam); anti-p-NMHC IIA (S1943) (1:2,000, Cell Signaling Technology); anti-MPO (1:1,000, Thermo Fisher Scientific); anti-Lactoferrin (1:500, Santa Cruz Biotechnology, Dallas, Texas, USA); anti-MMP9 (1:500, Santa Cruz Biotechnology); anti-mitofusin-2 (1:1,000, Cell Signaling Technology), anti-TOMM20 (1:1,000; Novus Biologicals); anti-VDAC (1:1,000; Merck Millipore); anti-OPA1 (1:1,000, Sigma-Aldrich); anti-Rho GDI (1:1,000, Santa Cruz Biotechnology); anti-actin (1:10,000, Cytoskeleton distributed by

LuBioScience GmbH) and anti-GAPDH (1:2,000; Merck Millipore). In the next morning, membranes were washed in TBST for 30 min at room temperature and incubated with the corresponding HRP-conjugated secondary antibody (GE Healthcare Life Sciences, Little Chalfont, UK) in TBST with 5% non-fat dry milk at room temperature for 1 h. After washing, membranes were developed with LI-COR Odyssey imaging system and signals were quantified using Image Studio software (LI-COR Biosciences, Lincoln, Nebraska, USA).

### Co-immunoprecipitation (Co-IP) assay

Neutrophils were lysed in lysis solution (25 mM Tris, 150 mM NaCl, 1 mM EDTA, 1% NP-40, 5% glycerol (pH 7.4)) provided by the Pierce IP Kit (Thermo Fisher Scientific), with addition of 100 μm PMSF, phosphatase inhibitors, and the freshly prepared protease inhibitor cocktail. Protein supernatants were obtained as described above and subjected to preclearing, using Pierce control agarose resin for 1 h at 4˚C. A total of 30 μg of protein from each sample were used as "Input." The immunoprecipitation procedure was performed following the IP kit instructions. Anti-HA antibody or anti-NMHC IIA antibody or the control antibody (Cat # X0936, Dako) was cross-linked to aminolink plus coupling resin using sodium cyanoborohydride for 2 h at room temperature. Immobilized antibody on the resin was washed with coupling buffer (10 mM sodium phosphate, 150 mM NaCl (pH 7.2)) and quenched in quenching buffer (1 M Tris-HCl). After further treatment with sodium cyanoborohydride, the antibody resin complex was thoroughly washed 6 times with wash solution. Thereafter, immobilized antibody resin was added to 1 mg precleared cell lysates and incubated overnight at 4˚C by rotating end-over-end. On the next day, the immune complexes were washed with lysis buffer and subjected to immunoblot analysis following elution. Specifically, to identify the potential interacting protein of RhoH, the elution products were separated by SDS-PAGE and then stained with Coomassie Brilliant Blue R-250 solution (Sigma-Aldrich). Thereafter, the bands of interest were excised from the gel and subjected to mass spectrometry analysis performed by Laboratory of Organic Chemistry, Department of Chemistry and Applied Biosciences, ETH Zurich, Switzerland.

### Mitochondria/cytosol fractionation assay

Subcellular fractionation was performed as previously described [70]. Briefly, neutrophils were lysed in cytosolic extraction buffer (CEB) (20 mM HEPES (pH 7.2), 250 mM sucrose, 10 mM KCl, 1.5 mM MgCl$_2$, and 2 mM EDTA) with digitonin (0.625 mg/ml) on ice for 20 min. Lysed cells were centrifuged at 700 g at 4˚C for 10 min. The supernatant was collected and centrifuged again at 7,000 g for 30 min at 4˚C. The supernatant was then transferred to a new tube for ultracentrifugation at 21,000 g at 4˚C for 60 min. The supernatant after ultracentrifugation contains the cytosolic fraction. The pellet after centrifugation at 7,000 g contains the mitochondrial fraction was washed 2 times with cold PBS. Thereafter, the cell pellet was lysed with CEB buffer containing 1% SDS by boiling at 95˚C for 5 min. The lysate was centrifuged at 14,000 rpm at 4˚C for 5 min. The supernatant after centrifugation at 14,000 rpm at 4˚C for 5 min that contained the mitochondrial fraction was collected for immunoblot analysis.

### Isolation of neutrophil granules

Neutrophil granules were isolated by density gradient ultracentrifugation using Lysosomal Enrichment Kit (Sigma-Aldrich) as previously described [47]. In brief, neutrophils were homogenized using a Dounce homogenizer and cell lysates were centrifuged at 500 g for 10 min to remove the nuclei. The postnuclear supernatant was layered onto a discontinuous Optiprep gradient (8% to 27%) and centrifuged at 150,000 g at 4˚C in a 50 TI fixed angle rotor in

Beckman Optima L-90K Ultracentrifuge. The 10 fractions were equally collected from top to bottom and diluted with cold PBS. Optiprep was removed by centrifugation of each fraction at 21,000 g for 30 min. The separated subsets were used for further analysis.

### Elastase activity assay

Elastase activity was determined using the EnzChek Elastase Assay Kit (Thermo Fisher Scientific) following the manufacturer's instructions. Fluorescence was measured with a spectrofluorometer (SpectraMax M2, Molecular Devices).

### Bacterial killing in vitro assay

A single colony of *E. coli* GFP (strain M655, a kind gift of E. Slack, ETH Zurich) was cultured in Luria broth base (LB) medium (Sigma-Aldrich) at 37˚C, shaking at 220 rpm overnight. The bacterial culture was diluted 1:100 in LB medium, grown to mid-logarithmic growth phase ($OD_{600} = 0.7$) and centrifuged at 1,000 g for 5 min. After washing twice with 1× HBSS, the bacterial pellets were gently centrifuged at 100 g for 5 min to remove the clumped bacteria. Bacteria were opsonized with 10% human IgG polyvalent (heat-inactivated) in 1× HBSS. Opsonized bacteria were mixed with separated fractions and rotated end-over-end for 30 min at 37˚C. The reaction was stopped by ice-cold 1× HBSS and samples were immediately analyzed by flow cytometry (FACSVerse, BD Biosciences).

### GTPase activity assay

Neutrophils were lysed and centrifuged at 14,000 g for 10 min at 4˚C. The supernatants were incubated with the GST-PBD (p21-binding domain of PAK1) beads for Rac1 and Cdc42 activity assays or GST-RBD (Rho-binding domain of Rhotekin) for RhoA activity assay following the manufacturer's instructions. Levels of bead-bound GTP-Rac1/Cdc42/RhoA and total Rac1/Cdc42/RhoA were analyzed by immunoblot with antibodies as follows: anti-Rac1 (1:1,000; Abcam), anti-Cdc42 (1:1,000; Abcam), anti-RhoA (1:1,000; Abcam).

### Actin-binding centrifugation assay

Purified G-actin was polymerized using F-actin polymerization buffer according to manufacturer's instructions (Cytoskeleton). Isolated granules and mitochondria were mixed and incubated for 30 min at room temperature in the presence or absence of 3.83 μm F-actin ex vivo. Actin bundling protein α-actinin was used as a positive control. Samples were then precipitated by centrifugation at 21,000 g, a force lower than needed to precipitate F-actin but suitable for precipitating granules and mitochondria. The precipitate and supernatant were evaluated by immunoblotting.

### Molecular docking analysis

The protein structures of RhoH and NMHC IIA were built by using SWISS-MODEL (https://swissmodel.expasy.org). The SWISS-MODEL template library (SMTL version 2020-08-05, PDB release 2020-07-31) was searched with BLAST and HHBlits for evolutionary related structures matching the target sequence. Models were built based on the target-template alignment using ProMod3. The best models referred to reported crystal structures of CDC42 (6SIU) and Myosin 2 heavy chain (3JAX) in the PDB database as templates were picked out for the prediction of the binding sites by performing ZDOCK SERVER (https://zdock.umassmed.edu). The top dock poses in the largest cluster were analyzed and visualized by PyMOL (https://pymol.org).

### *Escherichia coli* (*E. coli*)-induced peritonitis mouse model

*E. coli*-GFP were prepared as described above, and $50 \times 10^6$ *E. coli* in 150 μl PBS were injected intraperitoneally into mice to induce peritonitis as previously reported [54]. Serial dilutions of the final bacterial inoculum were distributed on agar plates and incubated overnight at 37°C to verify the number of viable bacteria injected. At designated time points after *E. coli* challenge, mice were anesthetized and sacrificed. The peritoneal cavity was flushed using 2 ml PBS containing 2.5 mM EDTA. The harvested peritoneal wash was distributed on agar plates in triplicate. After overnight incubation at 37°C, the number of colony-forming unit (CFU) was counted on the second day. The remaining peritoneal wash was centrifuged, and the supernatant was collected as PLF and aliquoted for further measurements. The peritoneal cells in the pellet were washed and subsequently stained with MitoSOX Red (5 μm) and 1 μg/ml Hoechst 33342 to visualize NETs or incubated with blocking buffer prior to cell surface staining with the following antibodies from BioLegend: PerCP-conjugated anti-CD45 and APC/cy7-conjugated anti-Ly6G antibodies. All samples were analyzed by flow cytometry (FACSVerse, BD Biosciences) followed by quantification using FlowJo software. Additionally, proteins were extracted from peritoneal cells and analyzed for RhoH protein expression by immunoblotting.

### Statistical analysis

Data were presented as means ± SD. Unpaired 2-tailed Student *t* test, 1-way or 2-way ANOVA was performed with GraphPad Prism software (San Diego, California, USA) as indicated in the figure legends. $P < 0.05$ was considered as statistically significant.

## Supporting information

**S1 Fig. *RHOH* expression is up-regulated in neutrophils under different inflammatory conditions.** (**A**) Heatmap showing fold change of *RHOH* gene expression in human neutrophils treated with GM-CSF at different time points (compared with freshly isolated neutrophils). The underlying data can be found in S1 Data. (**B**) Violin plots showing *RHOH* gene expression in cell populations identified in COVID-19 patients. Mono, monocytes; NK, natural killer cells; Macro, macrophages; Epi, epithelial cells; Neu, neutrophils, shown in black box (□); DC, dendritic cells; Plasma, plasma B cells; Mega, megakaryocytes. The underlying data can be found in https://www.ncbi.nlm.nih.gov/geo/query/acc.cgi?acc=GSE158055. (**C**) Violin plots showing the expression of *Rhoh* in major immune cell subsets identified in MC38 tumors. Neutrophils are shown in black box (□). The underlying data can be found in https://www.ebi.ac.uk/biostudies/arrayexpress/studies/E-MTAB-8832.
(DOCX)

**S2 Fig. Human neutrophils stimulated with serum from CF patients show similarity with CF neutrophils.** (**A**) Protein lysates of human neutrophils isolated from healthy donors treated with medium or serum from HD and CF patients (6 h) were analyzed by immunoblot for RHOH protein expression. (**B–D**) Pretreated human neutrophils with medium or serum from HD or CF patients (6 h) were subsequently activated with the indicated stimuli. (**B**) Neutrophil degranulation was determined using the surrogate marker CD63 by flow cytometry (sample number: 1, 2, 3, 4). (**C**) Quantification of released dsDNA in the culture supernatants (sample number: 1, 2, 3). (**D**) NETs were visualized by the colocalization of NE (green) with released dsDNA (PI, red) using confocal microscopy (sample number: 1, 2, 3). Scale bars, 10 μm. (**D**) Data are representative of 3 independent experiments. (**B, C**) Values are means ± SD. Two-way ANOVA with Šídák's multiple comparisons test was applied. The underlying data for S2B and S2C Fig can be found in S1 Data. The underlying data for S2A Fig can be found in

S1 Raw images. CF, cystic fibrosis; dsDNA, double-stranded DNA; HD, healthy donor; NE, neutrophil elastase; NET, neutrophil extracellular trap.
(DOCX)

**S3 Fig. Induction of RhoH expression by GM-CSF stimulation suppresses NET formation.** (**A**) Representative microscopic images of neutrophils freshly isolated from WT and *Rhoh*[-/-] mice. (**B**) The mRNA expression of *Rhoh* in WT and *Rhoh*[-/-] neutrophils was quantified by Q-PCR. (**C, D**) Freshly isolated neutrophils from WT and *Rhoh*[-/-] mice were pretreated with GM-CSF for indicated time followed by activation with LPS or PMA for 15 min. (**C**) Extracellular DNA fibers were stained with MitoSOX Red and the nucleus with Hoechst 33342 and analyzed by confocal microscopy. Scale bars, 10 μm. Data are representative of three independent experiments. (**D**) Quantification of released dsDNA in the culture supernatants. Unpaired 2-tailed Student *t* test (**B**) or 2-way ANOVA with Tukey's multiple comparisons test (**D**) was applied. Values are means ± SD. The underlying data for S3B and S3D Fig can be found in S1 Data. dsDNA, double-stranded DNA; GM-CSF, granulocyte/macrophage colony-stimulating factor; NET, neutrophil extracellular trap.
(DOCX)

**S4 Fig. HoxB8 neutrophils are characterized by nuclear morphology and flow cytometry.** (**A**) Representative images showing nuclear morphology of HoxB8 cells before and after differentiation into mature neutrophils. Scale bar, 10 μm. (**B**) Cell surface expression of Ly6G of HoxB8 cells before and after 5-day differentiation was analyzed by flow cytometry. Data are representative of 3 independent experiments.
(DOCX)

**S5 Fig. Reduced NMHC IIA expression by a second shRNA targeting *Myh9* decreases NET formation.** (**A**) Protein expression of NMHC IIA in mature HoxB8 neutrophils treated with control (Ctrl) or the second Myh9 shRNA was analyzed by immunoblotting. (**B**) Neutrophil degranulation was determined by flow cytometry. (**C**) Extracellular DNA fibers were analyzed by confocal microscopy. Scale bars, 10 μm. At least 3 independent experiments were performed. (**D**) Quantification of released dsDNA in culture supernatants. (B, D) Values are means ± SD. (B, D) Two-way ANOVA with Šídák's multiple comparisons test. The underlying data for S5B and S5D Fig can be found in S1 Data. The underlying data for S5A Fig can be found in S1 Raw images.
(DOCX)

**S6 Fig. Effects of myosin IIA inhibitors on human neutrophils.** (**A–C**) Human neutrophils were treated with vehicle control or myosin IIA inhibitors for 30 min followed by the indicated stimulation. (**A**) F-actin distribution was analyzed by confocal microscopy (upper part). Scale bars, 10 μm. Cell area and F-actin intensity were quantified by using Imaris software (lower part). A total of 24 images (each containing 8–15 cells) from 3 independent experiments were included for each condition. (**B**) Phosphorylation of myosin IIA was analyzed by immunoblotting. (**C**) ROS activity was assessed using DHR 123 dye by flow cytometry. Data are representative of 3 independent experiments. Values represent means ± SD. One-way ANOVA with Dunnett's multiple comparisons test was applied. The underlying data for S6A and S6C Fig can be found in S1 Data. The underlying data for S6B Fig can be found in S1 Raw images.
(DOCX)

**S7 Fig. Re-expression of RhoH does not affect the localization of NMHC IIA in neutrophil granules and mitochondria.** (**A**) The PNL from activated neutrophils expressing HA-RhoH or the EV was separated and collected equally into 1–10 fractions followed by immunoblot

analysis. (**B**) Immunoblot analysis of NMHC IIA in cytosolic (C) and mitochondrial (M) fractions from activated neutrophils expressing HA-RhoH or EV. Data are representative of 3 independent experiments. The underlying data for S7A and S7B Fig can be found in S1 Raw images. EV, empty vector; NMHC IIA, non-muscle myosin heavy chain IIA; PNL, postnuclear lysate.
(DOCX)

**S8 Fig. RhoH mutation at tyrosine 33 (RhoH$^{Y33F}$) results in lysosomal degradation of RhoH protein.** (**A**) The mRNA expression of *Rhoh* in neutrophils expressing WT or mutated HA-RhoH or the corresponding EV was quantified by Q-PCR. Values are means ± SD. One-way ANOVA with Dunnett's multiple comparisons test was applied. (**B**) Neutrophils expressing HA-RhoH$^{Y33F}$ were treated with MG132 or Baf A1 for 4 h followed by immunoblot analysis. MG132 (10 μm) was used as a proteasome inhibitor; Baf A1 (250 nM) was used as a lysosomal enzyme inhibitor. Rac1, Rho GDI, and actin were used as loading control. Values represent means ± SD from 3 independent experiments. One-way ANOVA with Tukey's multiple comparisons test was applied. The underlying numerical data for S8A and S8B Fig can be found in S1 Data. The uncropped immunoblots for S8B Fig can be found in S1 Raw images. EV, empty vector.
(DOCX)

**S9 Fig. RhoH is not involved in microtubule rearrangement and does not interfere with Cdc42 and RhoA activity.** (**A–D**) The 5-day differentiated HoxB8 neutrophils expressing HA-RhoH or EV were treated as indicated. (**A**) Microtubule assembly in these cells was analyzed by confocal microscopy (left). Scale bars, 10 μm. Quantification of microtubule was performed by automated analysis of microscopic images using Imaris software (right). Values are means ± SD. Two-way ANOVA with Tukey's multiple comparisons test was applied. (**B**) ROS activity was assessed by flow cytometry. (**C, D**) The levels of GTP-bound and total Cdc42 protein (**C**), GTP-bound and total RhoA protein (**D**), were compared using effector pulldown assay, followed by immunoblotting. GTPγS was used as a positive control, GDP as a negative control. All data are representative of 3 independent experiments. The underlying data for S9A and S9B Fig can be found in S1 Data. The underlying data for S9C and S9D Fig can be found in S1 Raw images. EV, empty vector.
(DOCX)

**S1 Data. The underlying numerical data for Figs 1C–1E, 1G, 2B, 2D, 2F, 2H, 3B, 3C–3E, 3H, 3I, 4C, 4D, 4F, 4G, 5A, 5C–5E, 5G, 6A–6D, 7B–7G and 7I, S1A, S2B, S2C, S3B, S3D, S5B, S5D, S6A, S6C, S8A, S8B, S9A and S9B Figs.**
(XLSX)

**S1 Raw images. The uncropped immunoblots for Figs 1B, 2E, 3A–3C, 3G, 4A, 4G, 5A, 5C, 5D, 6B–6D, 7A and 7H, S2A, S5A, S6B, S7A, S7B, S8B, S9C and S9D Figs.**
(PDF)

## Acknowledgments

We thank the participating patients and healthy blood donors for providing blood. We also thank the Laboratory of Organic Chemistry, Department of Chemistry and Applied Biosciences from ETH Zurich for performing LC-MS analysis. Images were acquired on equipment supported by the Microscopy Imaging Centre at the University of Bern. SP is a PhD student at the Graduate School of Cellular and Biomedical Sciences of the University of Bern.

## Author Contributions

**Conceptualization:** Shuang Peng, Shida Yousefi, Hans-Uwe Simon.

**Data curation:** Shuang Peng, Shida Yousefi, Hans-Uwe Simon.

**Formal analysis:** Shuang Peng.

**Funding acquisition:** Shida Yousefi, Hans-Uwe Simon.

**Investigation:** Shuang Peng, Darko Stojkov, Jian Gao, Kevin Oberson, Philipp Latzin, Carmen Casaulta.

**Methodology:** Shuang Peng, Darko Stojkov, Jian Gao, Kevin Oberson, Philipp Latzin, Carmen Casaulta, Shida Yousefi.

**Project administration:** Hans-Uwe Simon.

**Resources:** Hans-Uwe Simon.

**Software:** Jian Gao.

**Supervision:** Shida Yousefi, Hans-Uwe Simon.

**Validation:** Shida Yousefi.

**Visualization:** Shuang Peng.

**Writing – original draft:** Shuang Peng, Hans-Uwe Simon.

**Writing – review & editing:** Darko Stojkov, Jian Gao, Kevin Oberson, Philipp Latzin, Carmen Casaulta, Shida Yousefi, Hans-Uwe Simon.

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
