## [Editor Report · Decision Letter 0]

24 Feb 2022

Dear Uwe, 

Thank you for submitting your manuscript entitled "Nascent RHOH acts as a molecular brake on actomyosin-mediated effector functions of inflammatory neutrophils" for consideration as a Research Article by PLOS Biology. I have handled the manuscript in the absence of my colleague Richard Hodge this week. 

We have now discussed your manuscript with an academic editor of relevant expertise and I am happy to let you know that we would like to send your submission out for external peer review.

Once your full submission is complete, your paper will undergo a series of checks in preparation for peer review. Once your manuscript has passed the checks it will be sent out for review. To provide the metadata for your submission, please Login to Editorial Manager (https://www.editorialmanager.com/pbiology) within two working days, i.e. by Feb 26 2022 11:59PM.

If your manuscript has been previously reviewed at another journal, PLOS Biology is willing to work with those reviews in order to avoid re-starting the process. Submission of the previous reviews is entirely optional and our ability to use them effectively will depend on the willingness of the previous journal to confirm the content of the reports and share the reviewer identities. Please note that we reserve the right to invite additional reviewers if we consider that additional/independent reviewers are needed, although we aim to avoid this as far as possible. In our experience, working with previous reviews does save time. 

If you would like to send previous reviewer reports to us, please email me at rhodge@plos.org to let me know, including the name of the previous journal and the manuscript ID the study was given, as well as attaching a point-by-point response to reviewers that details how you have or plan to address the reviewers' concerns. 

Given the disruptions resulting from the ongoing COVID-19 pandemic, please expect some delays in the editorial process. We apologise in advance for any inconvenience caused and will do our best to minimize impact as far as possible.

Kind regards,

Richard

Richard Hodge

Associate Editor

PLOS Biology

rhodge@plos.org

---

## [Decision Letter · Decision Letter 1]

28 Mar 2022

Dear Uwe,

Thank you for submitting your manuscript "Nascent RHOH acts as a molecular brake on actomyosin-mediated effector functions of inflammatory neutrophils" for consideration as a Research Article at PLOS Biology. Your manuscript has been evaluated by the PLOS Biology editors, an Academic Editor with relevant expertise, and by several independent reviewers.

I have taken over the handling of your submission during Richard's absence from the office this week, in order to prvent any unnecessary loss of time. 

In light of the reviews (below my signature), although we will not be able to accept the current version of the manuscript, we would like to invite you to revise the work in response to the reviewer concerns. As you will see, reviewers 1 and 2 are very supportive of the work, despite raising some issues, and although reviewer 3 is more negative, the issues that are raised are addressable. However, in between the three reviewers, they do raise a substantial number of issues, and we have discussed with the Academic Editor which should be addressed for a successful revision, which should be discussed and which we would consider to be out of the scope of the present work. Detailed comments from the Academic Editor to this effect can be found at the end of the letter, below the reviewer reports. We think the summary of the feedback now amounts to what is reasonable to achieve during a revision.

Please note that we will not be able to make any decision about publication until we have seen the revised manuscript and your response to the reviewers' comments. Your revised manuscript is also likely to be sent for further evaluation by some or all of the reviewers.

We expect to receive your revised manuscript within 3 months. Please email us (plosbiology@plos.org) if you have any questions or concerns about the revision, or would like to request an extension. 

At this stage, your manuscript remains formally under active consideration at our journal; please notify us by email if you do not intend to submit a revision so that we may end consideration of the manuscript at PLOS Biology.

**IMPORTANT - SUBMITTING YOUR REVISION**

*Re-submission Checklist*

*Published Peer Review*

*PLOS Data Policy*

*Blot and Gel Data Policy*

With best wishes,

Nonia

Nonia Pariente, PhD

Editor in Chief

PLOS Biology

on behalf of 

Richard

Richard Hodge

Associate Editor

PLOS Biology

rhodge@plos.org

REVIEWS:

Reviewer #1: 

This paper identifies RhoH as a brake mechanism for neutrophil degranulation and NETosis. It shows that RhoH acts by interacting with the motor protein NMHC IIA, blocking the NMHC II-mediated association of organelles such as granules with the actin cytoskeleton, and thereby blocking degranulation and NETosis. The findings have great relevance for disease conditions such as cystic fibrosis, where neutrophils are recruited but unable to combat disease-associated bacterial infections. The paper is very well done, the authors have gone to great lengths to establish disease relevance and mechanistic insight.

Fig 1: RhoH, a constitutively active Rho GTPase is not expressed in normal neutrophils but upregulated during inflammation, in particular cystic fibrosis.

Fig 2: RhoH-/- mouse neutrophils show increased degranulation and NETosis. Re-expression of RhoH in neutrophil-differentiated Hoxb8 cells derived from RhoH-/- mice reduces degranulation and NETosis.

Fig 3: RhoH interacts with the motor protein NMHC IIA (aka myosin 9, Myh9) upon neutrophil stimulation with various agonists. Inhibition of NMHC IIA, or downregulation in Hoxb8 cells, reduces degranulation and NETosis.

Fig 4: NMHC IIA fractionates with granule and cytosol fractions. sh-RNA-mediated knockdown shows that NMHC IIA is required for the association of organelles (granules, mitochondria) with the actin cytoskeleton.

Fig 5: RhoH expression impairs F-actin formation and the activation of Rac1 in response to GM-CSF/C5a.

Fig 6: RhoH expression impairs the association of organelles with the F-actin cytoskeleton. Expression of a RhoH point mutant that cannot interact with NMHC IIA restores interaction of organelles with actin cytoskeleton, degranulation and NETosis.

Fig 7: Increased NETosis and killing of E. coli in RhoH-/- mice in a model of septic peritonitis.

Major comments:

1) The authors should the test RhoH in a mouse model of cystic fibrosis, in order to tie in the mouse data with the human patient data presented. Several mouse models of cystic fibrosis are available. Adoptive transfer of RhoH-deficient cells in such a mouse model would presumably alleviate CF-associated bacterial infections.

2) Figures 4G and 6D: please quantify the sedimented F-actin in treatment vs control samples, for all independent experiments and with statistics.

Minor comments:

3) Text to Fig 4B stating co-fractionation of NMHC IIA fwith granule markers: Please tone down, stating "partial" co-fractionation, as most of NMHC IIA is in the PNL fraction, shown to be cytosolic.

4) Fig 4B: Why show fractions 1-5 if they are empty of all markers and proteins investigated?

5) Labels of Figures 4G and 6D: why specify "buffer", all samples have buffer?

6) Text to Fig 5: Tone down interpretation that RhoH blocks actin polymerisation by inhibiting the activation of Rac1. This is not proven here, but correlated.

7) Discussion: In addition to the interaction of RhoH with NMHC IIA, could the reduced F-actin formation contribute to the impaired association of organelles with F-actin? Hard to imagine it would have no effect. Could the authors please comment in the discussion?

8) Discussion: could the authors please expand their discussion on how RhoH regulates Rac activity in neutrophils (e.g. as shown in ref 41)?

Reviewer #2: 

This is a very interesting paper identifying how neutrophil functions are impaired in chronic inflammation conditions such as in cystic fibrosis patients. The authors show that RHOH, a constantly active GTPase induced under inflammatory conditions, limits neutrophil degranulation and the formation of neutrophil extracellular traps (NETs). Neutrophils from CF patients display impaired ability to degranulation or produce NETs, accompanied with higher level of RHOH. RHOH is induced under inflammatory conditions in human neutrophils treated with serum from CF patients or GM-CSF. The authors further utilized a Rhoh-/- mice. GM-CSF and C5 stimulation reduced the effector function of WT murine neutrophils, but not the Rhoh -/- murine neutrophils. Furthermore, they reintroduce rhoh in the Rhoh -/- murine neutrophils and confirmed that the phenotype can be rescued. Over expressing Rhoh reduced neutrophil effector function. Finally, rhoh -/- mice has increased anti-microbial function. Together, the evidence that RHOH mediate the suppression of neutrophil function in inflammatory conditions are strong. To get to the mechanism, the authors showed that RhoH directly interact with NMHC IIA. Myosin IIA links neutrophil organelles to F-actin. A rhoh mutation that does not bind Myosin IIA does not inhibit neutrophil effector function as well as the WT. In parallel, RHOH inhibit the actin polymerization and reduce Rac1 activation. Whereas the biochemistry is relatively solid, the data presented does not fully support the conclusion and revision is necessary to address the weaknesses.

Major concerns:

1. RHOH suppresses neutrophil degranulation through two mechanisms. One is related to actin polymerization and the the other is through decoupling myosin-cargo from actin cable. It then raised the question: which mechanism is more important? Since the authors propose that traffic of the neutrophil organelle is dependent on F-actin, it seems that suppressing actin polymerization is a more fundamental mechanism of how rhoh operates. The authors used molecular docking and used the K34A mutation that disrupts binding to Myosin IIA. If the authors could show that K34A over expression does not disrupt the organization of the actin cytoskeleton, but only disrupts rhoh interaction with Myosin IIA, the biological significance of the cargo trafficking model can be supported.

2. Along the same line, can the authors rescue the knockout with a GDP-bound form of rhoh? Or overexpress this form?

3. The results in Figure 3 suggested that the interaction of rhoh and mysoin IIA is stimulated with GM-CSF+ C5a/LPS. The authors should discuss why this interaction is cell activation dependent. In addition, the immunoprecipitation is dependent on HA-RhoH overexpression. Will the authors be able to detect the rhph-myosin IIA interaction at the endogenous levels?

4. In figure 5, the authors observed reduced actin reorganization and reduced Rac1 activation in cells overexpressing HA-rhoH. Whereas the reduced F-actin can be explained with a reduced Rac1 activity, the results are not strong enough to suggest that rhoH regulate the actin cytoskeleton through Rac1 inhibition. There could be other mechanisms. In addition, in neutrophils, Rac2 is as important as Rac1. The authors should check Rac2 activation as well. The result is somewhat puzzling since Rac proteins are important for NADPH complex activation and ROS production. The authors should discuss how a significant reduction in Rac activation does not affect ROS production in RhoH over expressing cells.

5. The manuscript can be significantly strengthen to include live imaging of granules and mitochondria trafficking on F-actin in WT cell upon stimulation and how rhoh overexpression reduced cargo transport. Although the current data using biochemistry support the interaction of organelles with F-actin, whether the organelles travel along actin can only be validated with imaging in live cells.

Minor points,

1. In main text, results for figure 3a, bottom. 45.1% should be "coverage" rather than "identity". 

2. Flow graphs in Figure 1D,2B, 2F lack isotype control.

3. It would be helpful to quantify the SP ratio of actin in Figure 4G and 6A.

Reviewer #3: 

In this manuscript, the authors first investigate the role of RhoH in neutrophil function, from the starting point of cystic fibrosis-associated neutrophils. To find out how RhoH signals in neutrophils they immunoprecipitate HA-RhoH and identify NMH2A as an abundant protein in these immunoprecipitates, although without carrying out any relevant controls. They then investigate the role of NMH2A in neutrophil responses. They do not find any correlation between RhoH and NMH2A through a series of negative experimental results. They suggest that RhoH might interact directly with NMH2A but there is no biochemical analysis to show a direct interaction (using purified proteins in vitro) or any mutational analysis to identify sites involved in this proposed association. Without this information the results are mostly correlative without any direct causation links. Since RhoH is known to interact with PAKs, and PAKs can affect NMH2 activity, any effect of RhoH could be via PAK activation and thereby indirectly to NMH2A.

1. Abstract, p. 5 etc.: RhoH is not necessarily 'constitutively active'. It is 'constitutively GTP-bound' but there are multiple other ways to regulate Rho family members apart from GDP/GTP exchange via many different types of post-translational modifications. The wording should be altered to state specifically that it is 'constitutively GTP-bound'.

2. The Introduction should introduce what is known about RhoH in other cell types, especially in T-cells where it was first shown to inhibit Rac activation in 2002. It has been extensively studied for its role in T-cells, hematopoietic stem cells, B-cells and a variety of leukemias (including AML) in vivo and in vitro since then.

3. Introduction p. 5: do neutrophils express intermediate filament proteins (apart from lamins in the nucleus)? If intermediate filaments are not relevant to NET release then they should not be mentioned here - state specifically which cytoskeletal filaments are involved in NET release - is it microtubules, actin filaments or both?

4. Introduction: modify to state that 'RhoH is a member of the Rho GTPase family'.

5. Figure 1 and S2: are all the samples used in these figures from the same HD and CF patients? If so, please state which ones were used for each panel (referring to the numbering in the western blot, panel B). If not, the reason why not should be stated, and the patient samples numbered so that any common samples between experiments are clear to the reader. Presumably there are many transcriptional changes induced by CF serum in HD neutrophils apart from RhoH induction. Have the authors carried out a RNAseq experiment to determine how many genes are significantly changed? Do they know that RhoH induction is at the mRNA level or is it the protein that is stabilized following CF serum treatment? 

6. Figure 2: Is the induction of RhoH expression by GM-CSF at the mRNA level or protein stabilization? As for CF serum, presumably GM-CSF induces multiple changes in protein and mRNA expression - please provide background information on the number of changes. How does the level of HA-RhoH in Rhoh-/- neutrophils compare with the level of endogenous RhoH induced in wt neutrophils by GM-CSF? A western blot with both control and 180-min GM-CSF-treated HoxB-immortalized samples should be included for direct comparison of RhoH levels (wt, HA-RhoH/Rhoh-/- and Rhoh-/- cells).

7. Figure 3: How many proteins differ detectably between control and GM-CSF-treated cells? NMH2A is a highly abundant protein and highly abundant proteins are often detected non-specifically in immunoprecipitates. A control HA-tagged protein should be included to determine whether NMH2A is specifically pulled down with HA-RhoH or also with other proteins. The authors should also test whether known RhoH-interacting partners that are expressed in neutrophils (e.g. PAKs) are present in HA-RhoH immunoprecipitates, as a control for the immunoprecipitation. 

8. Figure 3: Only one shRNA targeting Myh9 was used - this means that the effects observed on neutrophil function could be due to an off-target effect. It is essential to test at least one different shRNA sequence targeting Myh9 to control for off-target effects.

9. Defining which region of RhoH and NMH2A are required for association and do they bind to each other directly?

10. Blebbistatin and ML7 do not specifically inhibit NMH2A but the activity of all myosin II proteins (NMH2A, 2B and 2C) so the text needs rewording. At the concentration of ML7 used here (30 µM), it would also inhibit PKA and PKC to some extent. Although it has been previously reported that NMH2B and NMH2C are not expressed in neutrophils (this should specifically be referred to in the Results text), the authors should check with the neutrophils they are using, particularly the HoxB8 cells. It is quite very likely that both B and C forms would be found in anti-HA immunoprecipitates, similar to NMH2A.

11. It would be expected that blebbistatin (and ML7) would induce spreading of unstimulated neutrophils, similar to what is observed with the inhibitors in other cell types (through relaxation of cortical actomyosin), rather than inhibit the spreading. The authors should test whether the two inhibitors affect the morphology of unstimulated neutrophils. The fact that ML7 alters the morphology of stimulated neutrophils suggests it could be acting on PKA/PKC for this effect. Alternatively, since blebbistatin is inactivated by blue light, it is possible that it was inactive in these experiments. In any case, the authors need to control for all these possibilities, as well as quantify effects of inhibitors on cell spread area over multiple randomly chosen cells from each experiment in at least 3 independent experiments. How many cells were analyzed in total and how many per experiment for the measurements of F-actin fluorescence (these numbers need to be added to the figure legend; note y-axis should say 'F-actin' not 'actin')?

12. Figure 5B and text: what do the authors mean by 'actin (should be F-actin) polarization'? They are not measuring F-actin polarization but simply total amount of F-actin fluorescence. There is no detail in the methods of how this F-actin quantification was carried out - this information should be added. If the authors aim to measure F-actin polarization, then they would need measure relative F-actin levels around the perimeter of each cell, correlating with cell shape factors (elongation, spread area etc.). As well as determining the effects of HA-RhoH expression, they should determine of rhoH depletion alters the F-actin distribution or cell shape in these HoxB8 cells.

13. The authors have not proven that HA-RhoH affects F-actin levels through Rac1 (p. 12), just that there is a correlation (and this is only shown for one experiment without quantification). This could only be proven by knocking down Rac1 (and Rac2 because it is more abundant than Rac1 in neutrophils). Testing the effects of rhoH depletion on Rac1/2 activity would be very informative, as in point 12, because if there is an effect it could be tested with the rhoH-knockout neutrophils in vivo.

14. The interpretation of the molecular docking analysis is overstated in the text. It can only suggest potential interaction sites not 'show' anything. There is also not enough information in the methods section to understand how the docking analysis was carried out. Since the structure of RhoH is not known, what Rho GTPase (PDB number) did they use to predict the RhoH conformation? And what did they use for modelling NMHC2A (PDB)? How could they dock RhoH onto NMHC2A? Did they use the structure of a known Rho GTPase structure with one of its known targets? It is not surprising that a charged residue (K34) is predicted to contribute to a model of the interaction, given that this is apparently not based on a known RhoH structure, and that charged residues are easy to 'dock'. It would be expected that adjacent residues such as Y33 would have some effect on any interaction, given that the 'effector loop' is linear in known Rho GTPase structures.

15. Important western blot results need to be quantified from the three independent experiments, to determine whether the changes are significant or not. For example, those shown in Fig. 6 using the K34A mutant need quantification, and Fig. S8 for the RhoH-Y33F mutant.

16. Absolute p values should be included for all experiments (including for results marked 'not significant') so that readers can judge for themselves whether changes are 'significant' or not. 

17. There are some grammatical errors throughout.

Comments from the Academic Editor

Rev 1

1. Outside the scope of the study

2. Should be addressed

3. Should be addressed

4. Should be addressed

5. Should be addressed

6. Should be addressed, related to point 8 of Rev 1, point 4 of Rev 2 and point 13 of Rev 3

7. Should be addressed, related to point 1 of Rev 2

Rev 2

1. Should be addressed

2. Should be addressed

3. Should be addressed

4. Should be addressed

5. Outside the scope of the study

6. Minor points should be addressed

Rev 3

1. Should be addressed

2. Should be addressed

3. No need to address

4. Should be addressed

5. Should be clarified, but no need to perform additional experiments

6. Should be addressed

7. Should be addressed

8. Should be addressed (see points 10,11)

9. Outside the scope of the study

10. 11. These two points are related to the specificity of Blebbistatin and ML7. I agree that including controls to make sure that blebbistatin is working as previously described is important. The effects of blebbistatin on naïve neutrophils have been reported (for example by the Weiner group, Cell 2012). Simple experiments describing the effects of the drug on neutrophil morphology should be included and compared to previously published data. The fact that KD of Myh9 gives similar results does provide confidence that the effects of blebbistatin are specific. Furthermore, the Myh9 KD findings also strongly suggest that the other two isoforms of NMH2 are not involved. However, in the absence of a rescue experiment, the requirement for a second shRNA sequence is valid.

12 Should be clarified, no experiments needed

13 Should be clarified

14 Should be clarified

15 16, 17 Should be addressed

---

## [Decision Letter · Decision Letter 2]

27 Jul 2022

Dear Uwe,

Thank you for your patience while we considered your revised manuscript "Nascent RHOH acts as a molecular brake on actomyosin-mediated effector functions of inflammatory neutrophils" for publication as a Research Article at PLOS Biology. Again, please accept my apologies for the delays that you have experienced during the peer review process. This revised version of your manuscript has been evaluated by the PLOS Biology editors, the Academic Editor and the original reviewers.

Based on the reviews, I am pleased to say that we are likely to accept this manuscript for publication, provided you satisfactorily address the remaining points raised by the reviewers. Whilst we appreciate that you have provided quantifications for the western blots presented in Figure 6, we ask that you please provide the quantifications and statistics in a graph format so that they are easier for the reader to spot. Please also make sure to address the following data and other policy-related requests that I have provided below (A-G):

(A) In the ethics statement provided in the manuscript:

- Please move the ethics statement to the Methods section (‘Mice’ subsection)

- Please provide the specific approval number issued by the Cantonal Veterinary Office of Bern and confirm whether the Veterinary office is an IACUC/ethics committee that consists of a panel of experts that reviewed and approved the study. In addition, please provide the method of euthanasia used in the experiments.

- Please provide the specific approval number issued by the Ethics committee of Canton Bern for the human subject’s research.

(B) Thank you very much for already providing the underlying data for the Figures presented in the manuscript. This looks complete, but I note that the underlying data for Figure S1B-C (GEO datasets) has not been included. I would be grateful if you could add this data in the source data file.

(C) For figures containing FACS data, we ask that you please submit the raw FCS files to the FlowRepository (https://flowrepository.org/). We also ask that this data be made publicly available in the repository before publication.

(D) Please also ensure that each of the relevant figure legends in your manuscript include information on *WHERE THE UNDERLYING DATA CAN BE FOUND*, and ensure your supplemental data file/s has a legend.

(E) In addition, thank you for already providing the original and uncropped images of the blots presented in the manuscript. However, I note that the images are not fully uncropped and do not show the wider area of the gel surrounding the bands. If you have cut the blot membranes during the analysis and do not have access to the whole gel, then this is OK. If you happen to have the wider image, then we ask that you please replace and include in the file. 

(F) Please ensure that your Data Statement in the submission system accurately describes where your data can be found and is in final format, as it will be published as written there. This would include providing the accession number/URL for any deposition at the FlowRepository. 

(G) Please also provide a blurb which (if accepted) will be included in our weekly and monthly Electronic Table of Contents, sent out to readers of PLOS Biology, and may be used to promote your article in social media. The blurb should be about 30-40 words long and is subject to editorial changes. It should, without exaggeration, entice people to read your manuscript. It should not be redundant with the title and should not contain acronyms or abbreviations. For examples, view our author guidelines: https://journals.plos.org/plosbiology/s/revising-your-manuscript#loc-blurb

We expect to receive your revised manuscript within two weeks. To submit your revision, please go to https://www.editorialmanager.com/pbiology/ and log in as an Author. Click the link labelled 'Submissions Needing Revision' to find your submission record. Your revised submission must include the following:

*Published Peer Review History*

*Press*

Best wishes,

Richard

Richard Hodge, PhD

Associate Editor, PLOS Biology

rhodge@plos.org

Reviewer remarks:

Reviewer #1: The original manuscript was very good, and the authors have addressed my comments in the revised version. They have toned down/removed previous overinterpretations, and I am very pleased that they have menasured Rac activity (new Figure 6). This showed that RhoH indeed negatively impacts on Rac1 activity, at least in a context of overexpression. I suspect that the RhoH deficient cells would also show a difference, with only slight tweaking of conditions, but I won't insist on that. I would also have preferred to see the Rac activity blots quantified by densitometry, but that alone does not warrant another round of revision. If other reviewers ask for a second round, please include this minor point.

Reviewer #2: The authors have addressed my concerns adequately.

Reviewer #3: The authors have addressed most of my points with additional data and/or changes to the text. 

Major concern: The new data provided (specifically the list of HA-RhoH interactors provided just for the reviewer) highlight that the interaction between RhoH and NMHCIIA is unlikely to be specific, as I stated in my original point 7. All of the proteins in the table are abundant and frequently detected by mass spectrometry in protein immunoprecipitations. Usually they are removed by proteomic bioinformaticians because they are known to be frequent contaminants, including myosins. I appreciate that the authors have some selectivity of RhoH/NMHCIIA association based on subsequent immunoprecipitations. Annexin-A1 is an equally plausible HA-RhoH 'interactor' based on the peptide coverage. Notably, a bioID interactome for RhoH did not detect NMHCIIA as an interactor (albeit in Hela and HEK293 cells, but these both express NMHCIIA). See Nature Cell Biol 2019, Cote JF (corresponding author). 

Note that some of my previous points have not been addressed:

1. I said that both the abstract and the introduction need to state that 'RhoH is constitutively GTP-bound'. The abstract still needs re-wording.

5. Please state specifically in the Introduction that RhoH protein was found to be upregulated by after CF treatment, and that RhoH mRNA was found to be upregulated by GM-CSF, as explained in their reply to my question.

11. Thank you for adding the cell spread area without GM-CSF. As expected based on my comments, blebbistatin alone induces cell spreading. Therefore it is not surprising that there is no additional effect of GM-CSF, because the cells are already spread. The text needs to be clear on this point. The p value for the difference of control cell spread area +/-blebbistatin should be included.

11. 'How many cells were analyzed in total and how many per experiment for the measurements of F-actin fluorescence?'. The reply says that Darko would provide these data but unfortunately the information is still missing from the figure legend.

13. The reply states 'Quantification has been performed and statistics were added (Figure 6C and 6D).' The quantification for the individual experiment is indicated, but there are no statistics (there should be results from 3 independent repeats quantified in a graph next to Figure 6C and Figure 6D). Also, I note that Rac2 is in fact activated by GM-CSF/C5a (based on the quantification of the single experiment shown). Quantification of multiple experiments is therefore essential, to prove whether this is reproducible or not.

14. The reply states' We agree with the Reviewer's point that the molecular docking analysis can only suggest potential interaction sites.' The text still says 'show' so needs changing. In addition, Supporting Figure 2D (alignment of RhoH with Rac1, Cdc42 etc.) is useful and should be added to the paper figures, as well as mentioned in the text.

15. Quantification of western blots: Figure 6D has not been quantified, even though it is in the authors' list of 'quantified blots', and needs to be done. Rac1/Rac2 activity assays can be very variable between experiments, hence the need to show reproducibility.

---

## [Editor Report · Decision Letter 3]

11 Aug 2022

Dear Uwe,

On behalf of my colleagues and the Academic Editor, Carole Parent, I am pleased to say that we can accept your manuscript for publication, provided you address any remaining formatting and reporting issues. These will be detailed in an email you should receive within 2-3 business days from our colleagues in the journal operations team; no action is required from you until then. Please note that we will not be able to formally accept your manuscript and schedule it for publication until you have completed any requested changes.

PRESS

Thank you again for thinking of us for your manuscript and supporting Open Access publishing. We are looking forward to publishing your study. 

Best wishes, 

Richard Hodge, PhD

Associate Editor, PLOS Biology

rhodge@plos.org

PLOS
